# Chasing the Tail: Effective Rubric-based Reward Modeling for Large Language Model Post-Training

**Junkai Zhang**[*][†]
University of California, Los Angeles

**Zihao Wang**[*]
Scale AI, Inc.

**Lin Gui**[*]
University of Chicago

**Swarnashree Mysore Sathyendra**
Scale AI, Inc.

**Jaehwan Jeong**
Scale AI, Inc.

**Victor Veitch**
University of Chicago

**Wei Wang**
University of California, Los Angeles

**Yunzhong He**
Scale AI, Inc.

**Bing Liu**
Scale AI, Inc.

**Lifeng Jin**
Scale AI, Inc.

## Abstract

Reinforcement fine-tuning (RFT) often suffers from *reward over-optimization*, where a policy model hacks the reward signals to achieve high scores while producing low-quality outputs. Our theoretical analysis shows that the key lies in reward misspecification at the high-reward tail: the inability to reliably distinguish *excellent* responses from merely *great* ones. This motivate us to focus on the high-reward region. However, such tail examples are scarce under the base LLM. While off-policy exemplars (e.g. from stronger models or rewrites) are easier to obtain, naively training on them yields a misspecified reward for the policy we aim to align. To address this, we study *rubric-based rewards*. By design, rubrics can leverage off-policy examples while remaining insensitive to their artifacts. To elicit rubrics that capture the high-reward tail, we highlight the importance of distinguishing among **great** and **diverse** responses, and introduce a workflow to implement this idea. We empirically demonstrate that rubric-based rewards substantially mitigate reward over-optimization and deliver effective LLM post-training improvements.[*]

## 1 Introduction

In this paper, we are interested in how to produce reward models that are effective when used for LLM post-training. A reward model is a function that takes a prompt and a response and produces a score quantifying how good that response is for the prompt. In post-training, we then align a language model to the reward by a reinforcement learning type procedure. The fundamental challenge here is that, in many settings, it is nearly inevitable that the reward model will be an imperfect proxy for the behavior that we are actually trying to induce. In particular, this means that as we run post-training, it will increasingly be the case that the LLM is aligned to the idiosyncratic misspecification of the reward rather than the true signal that we are trying to extract. In this paper, we are interested in mitigating this effect.

Given that some misspecification is inevitable, what should we focus on when defining a reward model? The basic setup of post-training aims to induce the good behavior encoded by the reward while minimally shifting other aspects of the base LLM. Mathematically, this can be formalized

---

[*]Equal contribution.
[†]Work done during internship at Scale AI.
[*]Our code can be accessed at https://github.com/Jun-Kai-Zhang/rubrics.git.

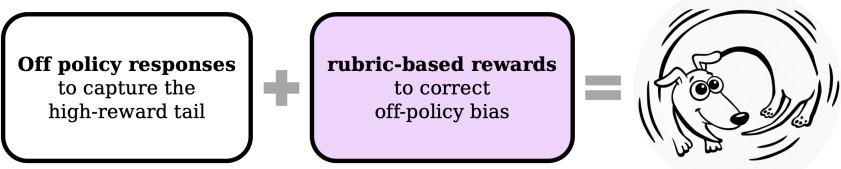

Figure 1: Chasing the Tail with Rubric-Based Rewards

as looking for post-training procedures that move along the Pareto frontier of KL divergence from the base model vs win rate (as judged by the reward) against the base model. We begin by theoretically demonstrating that, for such Pareto-optimal procedures, the effect of reward misspecification is dominated by errors in the high-reward region. In other words, what really matters for post-training is the ability to accurately distinguish between the very good responses.

Then, we know that we want to focus our reward modeling on the high-reward region of examples. The basic challenge here is that actually producing high-reward examples to train a reward model on is hard. If we simply sample responses from the base LLM itself, then it is extremely sample inefficient to get the necessary examples (because we are trying to get elements in a low-probability tail). On the other hand, if we use an off-policy procedure—e.g., drawing samples from a stronger LLM, or producing good examples with extensive thinking or rewrites—we can get high-reward examples, but naively training a reward model on them may learn superficial features instead of eliciting real capabilities (see Appendix D).

To address this challenge, we empirically study *rubric-based rewards* as a solution to this problem. In essence: we get very strong examplar responses by using off-policy generation. Then, we produce a reward model using these examples by using another LLM to produce a grading rubric for each prompt. Such rubric-based rewards will generalize well off-policy because they are insensitive to irrelevant aspects of the responses by design. The question is then if, and how, we can elicit rubrics that succeed in capturing the high-reward tail behavior necessary for alignment. We give two principles for achieving this goal. We then produce a workflow implementing these ideas and show empirically that it is highly effective for the LLM post-training task.

Summarizing, the contributions of this paper are:

1. A theoretical characterization of *how* reward misspecification matters for post-training, concluding that the high-reward region is key,

2. A method for constructing effective reward rubrics using off-policy data, and

3. An empirical study showing the efficacy of the constructed rubrics for post-training, and confirming the critical role of misspecification in the high-reward region.

## 2 PRELIMINARIES

**Notations.** We use $\pi$ to denote a large language model (LLM) and $\pi_0$ to denote the reference language model (usually the starting point of RL). Given a prompt $x$, a response $y$ is sampled from the conditional distribution $\pi(\cdot \mid x)$. A reward model $r(\cdot, \cdot)$ is utilized to assess the quality of a prompt-response pair. We use $r^\star$ to represent the gold reward model (inaccessible in practice) and $r$ to represent the proxy reward applied in practice.

**Reinforcement fine-tuning (RFT).** With a prompt set $D$ and a reward model $r$, the reinforcement fine-tuning optimizes the following objective (Ouyang et al., 2022; Bai et al., 2022):

$$\max_{\pi} \mathbb{E}_{x\sim D,\ y\sim\pi(\cdot|x)}\left[r(x,y)\right] - \beta\mathbb{D}_{\mathrm{KL}}\left[\pi(y\mid x)\|\pi_0(y\mid x)\right], \quad (1)$$

where $\beta$ is a hyperparameter to control fine-tuned model's deviation from the reference model, i.e.,

$$\mathbb{D}_{\mathrm{KL}}\left[\pi(y\mid x)\|\pi_0(y\mid x)\right] = \mathbb{E}_{x\sim D,y\sim\pi(\cdot|x)}\left[\log\frac{\pi(y\mid x)}{\pi_0(y\mid x)}\right].$$

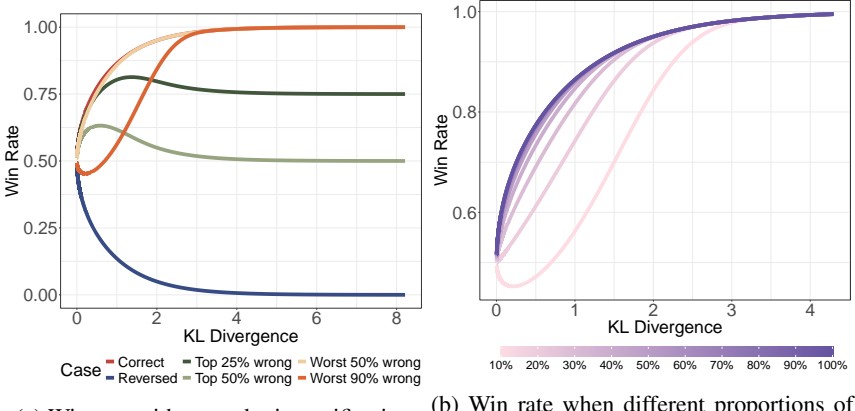

(a) Win rate with reward misspecification

(b) Win rate when different proportions of top responses are correctly ranked

Figure 2: Theoretical impact of reward model misspecification on performance. (a) Inaccuracy in the high-value region causes performance to collapse. (b) Correctly ranking top responses is sufficient for near-optimal performance.

As demonstrated in Rafailov et al. (2023), the solution to (1) is

$$\pi_r(y \mid x) \propto \pi_0(y \mid x) \exp\{r(x, y)/\beta\}. \tag{2}$$

**Reward over-optimization.** Because RFT relies on proxy rewards in practice, it inevitably suffers from *reward over-optimization*: the policy exploits inaccuracies in the reward model, achieving high proxy scores while true quality deteriorates. This phenomenon has been well studied in Bradley-Terry reward models trained on human preference data (Gao et al., 2023). The standard remedy is online RLHF, where fresh human feedback is periodically collected to update the reward model and mitigate over-optimization (Bai et al., 2022), but such approaches are costly and slow.

**Reinforcement learning from rubrics-based reward.** Reinforcement learning from rubrics-based reward (RLRR) (Gunjal et al., 2025; Viswanathan et al., 2025; Huang et al., 2025b) has emerged as a promising approach for open-ended tasks. The core idea is to associate each prompt $x$ with a rubric—a set of explicit criteria $(c_i)$ with corresponding weights $(w_i)$ that collectively define a high-quality response. For instance, given a prompt asking for a likely diagnosis from a patient's symptoms, the rubric could specify key aspects of a good answer. This might include high-weight criteria for "identifying [correct diagnosis] as the likely diagnosis" and "correctly identifying the condition as a medical emergency," and a low-weight criterion for "mentioning [typical treatment] for treatment" (See Appendix I for a concrete example.)

In this framework, a verifier $V$, typically another LLM, assesses whether a given response $y$ satisfies each individual criterion. The total reward is then calculated as the weighted average of the criteria that the response successfully meets. Formally, the verifier outputs a binary score for each criterion, $V(x, y, c_i) \mapsto \{0, 1\}$, and the total reward is:

$$r(x, y) = \frac{\sum_i w_i V(x, y, c_i)}{\sum_i w_i}.$$

RLRR extends Reinforcement Learning with Verifiable Rewards (RLVR) to general tasks where performance cannot be easily verified. Compared to RFT using Bradley-Terry reward models, RLRR's explicit criteria make rewards more interpretable and harder to game. However, it's still unclear if, and how, RLRR alleviates reward over-optimization.

## 3 HIGH-REWARD REGION ACCURACY IS KEY TO OVERCOMING REWARD OVER-OPTIMIZATION

It's well-known that using misspecified proxy rewards lead to reward over-optimization for reinforcement post-training. However, the ways in which different misspecification patterns of proxy

rewards influence the performance of the aligned model remain poorly understood. In this section, we develop theoretical results showing that maintaining high-reward region accuracy is the key determinant of alignment quality.

We introduce a *misspecification mapping* $f$ from gold to proxy rewards and cast the problem as analyzing how the geometry of $f$ affects performance. More specifically, $f : \mathbb{R} \to \mathbb{R}$ is the mapping from $r^\star$ to $r$, i.e., for any $x$-$y$ pair,

$$f\left(r^\star(x, y)\right) = r(x, y).$$

To characterize the reward over-optimization phenomenon, we need to study the relationship between the utility (expected reward and win rates), and the KL divergence in (2). They can be simplified as follows:

**Proposition 1.** *Define $R_0^x = r^\star(x, Y_0)$ with $Y_0 \sim \pi_0(\cdot \mid x)$ and $F_0^x$ as its cumulative distribution function. The RFT solution* (2) *has:*

*(i) Expected reward:* $\mathbb{E}_{x \sim D,\, y \sim \pi_r(\cdot \mid x)}\left[r^\star(x, y)\right] = \mathbb{E}_{x \sim D}\left[\dfrac{\mathbb{E}\left[R_0^x\, e^{f(R_0^x)/\beta}\right]}{\mathbb{E}\left[e^{f(R_0^x)/\beta}\right]}\right],$

*(ii) Win Rate:* $\mathbb{E}_{x \sim D,\, y \sim \pi_r(\cdot \mid x)}\left[F_0^x\left(r^\star(x, y)\right)\right] = \mathbb{E}_{x \sim D}\left[\dfrac{\mathbb{E}\left[F_0^x(R_0^x)\, e^{f(R_0^x)/\beta}\right]}{\mathbb{E}\left[e^{f(R_0^x)/\beta}\right]}\right],$

*(iii) KL divergence:* $\mathbb{D}_{\mathrm{KL}}\left[\pi_r(y \mid x) \| \pi_0(y \mid x)\right] = \mathbb{E}_{x \sim D}\left[\dfrac{\mathbb{E}\left[f(R_0^x)\, e^{f(R_0^x)/\beta}/\beta\right]}{\mathbb{E}\left[e^{f(R_0^x)/\beta}\right]} - \log \mathbb{E}\left[e^{f(R_0^x)/\beta}\right]\right]$

To proceed, we assume the current policy's ground-truth reward, $R_0^x$, is distributed from the standard uniform. This assumption is valid since: (i) it matches the reward distribution of best-of-n sampling and the optimal solution which best balance KL divergence and win rate (Gui et al., 2024; Azar et al., 2024; Balashankar et al., 2024) , and (ii) win rate and expected reward matches each other in this case. Under this assumption, we can characterize the utility-KL tradeoff when applying the misspecifed rewards:

**Theorem 1.** *Suppose each $R_0^x \sim U(0, 1)$ and $f(R_0^x) \stackrel{d}{=} R_0^x$. Then it holds that:*

*(i) KL divergence is invariant to $f$:*

$$\mathbb{D}_{\mathrm{KL}}\left[\pi_r(y \mid x) \| \pi_0(y \mid x)\right] = \frac{(1/\beta - 1)e^{1/\beta} + 1}{e^{1/\beta} - 1} - \log \beta - \log(e^{1/\beta} - 1).$$

*(ii) Expected reward (or win rate) of $\pi_r$ is* $\dfrac{\int_0^1 f^{-1}(u) e^{u/\beta}\, \mathrm{d}u}{\beta\left(e^{1/\beta} - 1\right)}$. *[Proof].*

The explicit formula in Theorem 1 indicates that misspecification, i.e., the deviation of $f$ from the identity map, in the high-value region of $r^\star$ has dominantly large effects on the utility-KL tradeoff. On one hand, the KL divergence remains invariant to the choice of $f$ and is fixed when the penalty parameter $\beta$ is set. On the other hand, the exponential term imposes increasingly severe penalties on misspecification in the high-reward regime relative to the low-reward regime. This highlights the criticality of accuracy in the high-reward region for achieving a favorable balance between utility and KL divergence.

To verify this, we investigate different $f$s and exactly compute the utility-KL tradeoff curves:

(i) "Correct": identity mapping $f(r^\star) = r^\star$

(ii) "Reversed": the reverse mapping $f(r^\star) = 1 - r^\star$

(iii) " Top $c\%$ wrong": $r = f(r^\star) = r^\star 1_{\{r^\star \leq 1-c\}} + (2 - c - r^\star)1_{\{r^\star > 1-c\}}$, i.e., the proxy reward model provides completely reverse rewards for highest quality responses

(iv) "Worst $c\%$ wrong": $r = f(r^\star) = (c - r^\star)1_{\{r^\star \leq c\}} + r^\star 1_{\{r^\star > c\}}$, i.e., the proxy reward model provides completely reverse rewards for worst quality responses

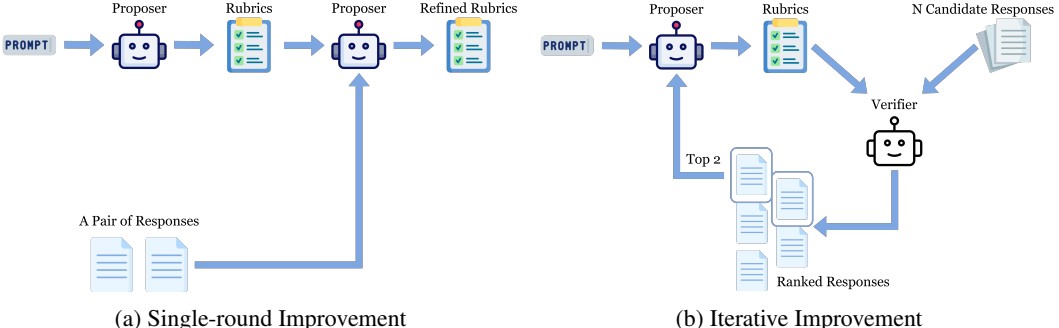

Figure 3: Rubric refinement through response differentiation. (a) Single-round: A proposer LLM analyzes a pair of responses to identify distinguishing features and encodes them as new rubric criteria. (b) Iterative: Multiple rounds progressively focus on higher-quality responses, with each iteration filtering to top-scoring candidates before generating new differentiating rubrics.

Figure 2a plots KL divergence versus win rate across misspecification patterns and yields two key observations: (i) when the proxy is inaccurate in the *high-reward* region, performance may look acceptable at small KL but the win rate collapses as KL grows (this is similar to the reward over-optimization behavior in Gao et al. (2023)); and (ii) if the proxy correctly ranks just a small top proportion of responses (e.g., $10\%$), even while misgrading the remaining majority, the win rate rapidly approaches the optimal curve at moderate KL. Separately, Figure 2b varies the fraction $c$ of correctly ranked top responses and traces the corresponding lower envelope of achievable win rates, showing that this envelope is already near-optimal once a sufficiently large top proportion is correctly identified and ordered (e.g., $40\%$). Together, we reach our central theoretical findings:

( I ) *Reward over-optimization primarily arises from the inaccuracy in high-reward regions.*

( II ) *Being able to accurately rank and differentiate high-quality outputs is sufficient for a reward model to effectively guide RL.*

---

**Algorithm 1** Iterative Rubric Refinement through Progressive Differentiation

---

1: **Input:** Pool of candidate responses and initial rubrics
2: **Iteration:** For each refinement round:
   (a) Score all candidate responses with the current rubrics and get the top 2 responses from the candidate pool as the comparison pair.
   (b) Use the proposer LLM to identify distinguishing features between the pair and encode these features by refining the existing rubric set.
3: **Output:** Final refined rubric set

---

## 4 PRINCIPLES FOR CONSTRUCTING RUBRICS

Based on the results of the previous section, we construct a reward model focusing on the high-value region. The problem then is getting training examples that are in this high-reward region. By definition, these are samples that are rare under the base LLM policy! This essentially forces us to use off-policy data to define the reward model. Now, *rubric-based rewards* have emerged as an approach for using off-policy data to define rewards. The basic idea of rubric-based reward models is to explicitly restrict the reward to only care about aspects of the solution that are relevant to its quality, thereby mitigating the side effect of the off-policy data. However, the restrictive nature of the rubrics is a double-edged sword. The same structure that limits the effect of off-policyness may also limit their ability to distinguish between solutions that are *excellent* and those that are merely *great* (they can easily end up in a tie). In this section, we consider how to construct rubrics that are focused on accuracy in the high-reward region.

Table 1: RL experimental results across three domains. The base policy model is Qwen3-8B-Base, and the win rate is evaluated against Qwen3-8B. The results show two clear trends: refining rubrics by differentiating two *great* responses outperforms differentiating two *good* responses, and differentiating among a *diverse* set of *great* responses further improves performance.

| Method | Generalist Domain | | Health Domain | | Finance Domain | |
|---|---|---|---|---|---|---|
| | Filtered Set | LMArena | Medical-o1 | | Finance | |
| | Win% | Win% | Win% | Score | Win % | Score |
| Base Policy | 5.2 | 4.1 | 10.8 | 0.1721 | 5.8 | 0.1738 |
| SFT | 35.9 | 29.6 | 25.8 | 0.2999 | 26.0 | 0.2218 |
| Initial, Prompt only | 31.3 | 29.7 | 21.7 | 0.3004 | 37.2 | 0.2693 |
| 1 *good* Pair | 33.5 | 32.8 | 22.4 | 0.2912 | 39.1 | 0.2694 |
| 1 *great* Pair | **36.8** | **33.1** | **26.5** | **0.3163** | **42.2** | **0.2838** |
| 4 *great* Pairs | 38.7 | 34.7 | 31.4 | 0.3348 | 48.9 | 0.2961 |
| 4 *great* & *Diverse* Pairs | **39.7** | **35.1** | **34.4** | **0.3513** | **49.6** | **0.3018** |

Refining rubrics to reliably tell apart two already *great* responses is a natural first step toward capturing the high-reward tail. To push accuracy further in that tail, we also update rubrics to distinguish among a *diverse* set of *great* responses. We formalize these ideas as two principles for rubric construction

---

**Principles for Rubric Construction**

```
[Principle 1]  Effective rubric construction requires
               distinguishing excellent responses from great
               ones.
[Principle 2]  Effective rubric construction requires
               distinguishing among diverse off-policy
               responses.
```

---

## 4.1 METHODOLOGY

To operationalize the above principles, we design an iterative workflow that leverages off-policy responses to refine rubrics.

**Refinement-through-Differentiation (RTD).** A natural way to make rubric-rewards more discriminative is to prompt a proposer LLM with a pair of *candidate responses* and the current rubrics. The proposer analyzes the pair, identifies their distinguishing features, and encodes these distinctions as new rubric criteria or refinements of existing ones. We refer to this fundamental refinement step as *Refinement-through-Differentiation* (RTD).

**Iterative workflow for chasing the tail.** While a single RTD step sharpens the rubric, repeated application over a larger candidate pool yields systematic improvements. Starting with all off-policy responses for a prompt, each iteration scores the candidates under the current rubric, selects the top two responses, and refines the rubric using RTD. This workflow concentrates rubric discovery on the performance frontier, extracting the most informative distinctions from the best available responses with only a small number of comparisons (see Algorithm 1 and Figure 3).

## 4.2 EXPERIMENTAL SETUP

Our experimental goals are twofold. First, we examine how leveraging off-policy responses can alleviate reward over-optimization. Second, we assess the efficacy of these methods in enhancing LLM capabilities. Our experiments span three domains: general-purpose, healthcare, and finance. For the

first goal, to isolate the effect of rubric refinement strategies, we adopt the synthetic oracle setting proposed by the seminal work Gao et al. (2023). In this framework, a strong LLM acts as a proxy for the "gold-standard" preference source. This oracle model serves a dual purpose: it generates the rubrics data used for reward modeling and serves as the final judge for performance evaluation. By unifying the annotation and evaluation sources, this design eliminates confounding factors arising from preference mismatch between the training data and the judge. This allows us to clearly assess how effectively different rubric refinement strategies mitigate reward over-optimization. For the second goal, we evaluate the model performance on domain-specific objective benchmarks when applicable. Regarding the second goal, we specifically evaluate the model on healthcare and finance tasks. These fields offer established objective professional benchmarks, allowing us to rigorously test improvements in model capabilities. We set up the experiments as follows:

**Training setup.** We employ GPT-4.1 as the *rubric proposer*, prompting it to generate the *initial rubrics*. The *training datasets* consist of two generalist prompt collections (LMArena (Chiang et al., 2024) and a manually filtered set of natural prompts, detailed in Appendix G) and two domain-specific prompt sets: for healthcare, we utilize medical-o1-reasoning-SFT (Chen et al., 2024); for the finance domain, we filtered for 1147 high-quality finance prompts in LMArena Team (2025). Each dataset contributes 5000 prompts for training and an additional 1000 prompts for in-domain evaluation, with the exception of the finance dataset, for which we use all available prompts for training and the prompts from the PRBench-Finance (Akyürek et al., 2025) for the evaluation. The *base model* for post-training is Qwen3-8B-Base (Yang et al., 2025), which has instruction-following capabilities. We adopt GRPO (Shao et al., 2024) as the RFT algorithm and use a standard set of hyperparameters, detailed in Table 4. For the reward computation, we leverage GPT-4.1-mini as a *rubric verifier* and calculate the final reward as the weighted sum of satisfied rubric criteria, normalized by the total weight. All prompts used in the experiments are presented in Appendix B.

**Candidate pool.** To validate `Principle 1`, we compare rubrics refined using (i) candidate pairs from a *great* model versus (ii) candidate pairs from a *good* model (Gemini 2.5 Pro and Gemini-2.5-Flash-Lite, respectively (Comanici et al., 2025)). To validate `Principle 2`, we enlarge the pool by sampling 16 responses per prompt, from a broader set of *excellent* models, ensuring greater diversity (see Appendix F for the full list). This setup allows us to test whether rubric refinement benefits from better and more diverse candidate responses. The supervised fine-tuning (SFT) uses two responses per prompt, sampled from the *great* model.

**Evaluation.** We assess performance across two primary dimensions: (1) alignment with oracle preferences, and (2) domain-specific scores on professional benchmarks. For preference evaluation, we conduct head-to-head comparisons against Qwen3-8B, the strong thinking version of our base policy model, on a held-out set of test prompts. The oracle model GPT-4.1 was prompted to act as an impartial evaluator and select the better response (see Appendix E for the detail). To validate the performance gain in professional domains, we additionally evaluate models on HealthBench (Arora et al., 2025) and PRBench (Akyürek et al., 2025), which provide objective metrics grounded in expert-curated rubrics.

## 5 RESULTS

### 5.1 RL IMPROVES WITH BETTER AND MORE DIVERSE RESPONSES

We first evaluate downstream RL performance to test whether the proposed principles indeed improve rubrics. Table 1 shows two clear trends. First, rubrics refined with *great* pairs outperform those refined with *good* pairs, validating `Principle 1`. Second, iterative refinement with multiple diverse *great* pairs yields further gains, validating `Principle 2`.

Beyond improving average performance, refinement with better and more diverse responses also mitigates reward over-optimization. Figure 4 shows training dynamics on the health and finance domains when RL is run for extended steps. Models trained on initial rubrics, or rubrics refined with only a single pair, peak early and then suffer a rapid decline in win rate and benchmark scores—an indicator of reward over-optimization. In contrast, models trained with iteratively refined, diverse rubrics sustain higher win rates and benchmark scores for much longer, with over-

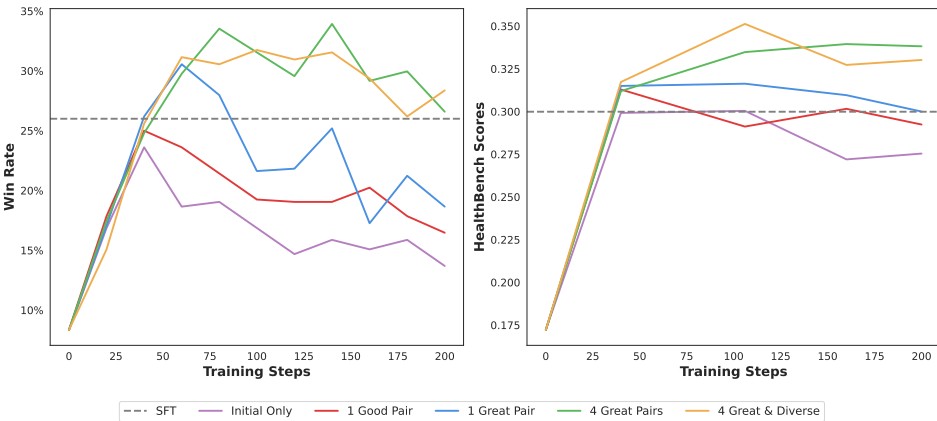

(a) Healthcare: Win Rate and HealthBench Scores at Different Training Steps

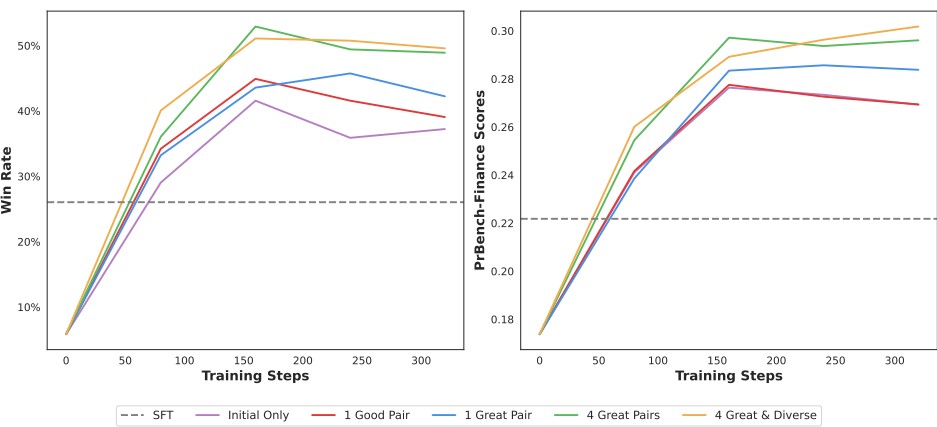

(b) Finance: Win Rate and PRBench-Finance Scores at Different Training Steps

Figure 4: Training dynamics under different rubric construction strategies over separate prolonged training. The figures show the evolution of the Win Rate and respective benchmark scores in the healthcare (a) and finance (b) domains.

optimization not appearing until late stages. This pattern indicates that refining rubrics with *great* and *diverse* responses corrects inaccuracies in the high-reward region, thereby delaying the onset of over-optimization. Together, these results confirm our central hypothesis that rubrics can be constructed to mitigate reward over-optimization.

## 5.2 REWARD MODEL ACCURACY IMPROVES IN THE HIGH-REWARD TAIL

Our theoretical analysis (Section 3) suggests that accuracy in the high-reward tail is the critical factor for downstream RL performance. To understand why refinement with better and more diverse responses helps, we evaluate the agreement between rubric-based rewards and the ground-truth judge, separately on the high- and low-reward regions.

As shown in Table 2, incorporating any candidate responses through refinement improves rubric accuracy compared to the prompt-only baseline. More importantly, rubrics refined with *great* pairs largely improve accuracy in the high-reward region, while *good* pairs improve accuracy more than *great* pairs in the low-reward region. Iterative refinement with *great* pairs pushes the accuracy in the high-reward region even further, mirroring the RL improvements in Table 1. This confirms that both principles work by sharpening reward model accuracy where it matters most: the high-value tail.

Table 2: Accuracy of rubric-based scoring in predicting ground-truth model preferences was evaluated on 1000 random prompts from the training set. Response pairs in the high-reward region were sampled from Qwen3-8B, and response pairs in the low-reward region were sampled from Qwen3-8B-Base. Rubric preferences were determined by a majority vote from five independent gradings, **with ties counted as incorrect**. Results show refining with stronger and more diverse responses improves high-reward accuracy.

|  | Initial Only | 1 *Good* Pair | 1 *Great* Pair | 4 *Great* Pairs | 4 *Great* & *Diverse* Pairs |
|---|---|---|---|---|---|
| High-reward | 40.3% | 42.2% | **45.8%** | **49.2%** | 47.9% |
| Low-reward | 66.2% | **67.9%** | 66.7% | 68.9% | **69.8%** |

## 5.3 REFINEMENTS FROM BETTER RESPONSES ARE MORE SOPHISTICATED

Finally, we analyze how refinements differ when using *good* versus *great* candidate responses. To understand how stronger candidate responses lead to better rubrics, we analyzed the types of refinements made when using different quality levels of candidate pairs. We prompted an LLM to compare initial and refined rubrics, and categorized the improvements into semantic clusters (see details in Appendix H).

Table 3 shows the distribution of refinement types on the health domain. Both qualities contribute, but the patterns diverge: *good* responses often drive **basic corrections**, such as adding penalties for obvious mistakes or broadening overly restrictive criteria; by contrast, *great* responses more often drive **sophisticated refinements**, such as breaking down complex criteria into sub-components or enhancing verification standards.

In the example from Appendix I, for a medical prompt about a patient with serious symptoms, two initially tied *great* responses are distinguished by adding the criterion: "The response mentions that urgent imaging (e.g., contrast-enhanced CT or MRI/MRV) is required to confirm the diagnosis." This refinement, from the "Enhancing verification, validation, and evidence standards" cluster, mandates a critical, verifiable clinical action, and only one of the responses satisfies. Such qualitative results confirm our finding that comparing *great* responses provides the nuanced distinctions needed to identify *excellent* outputs, thereby sharpening accuracy in the high-reward tail.

Table 3: Distribution of rubric refinement types when using *great* (Gemini 2.5 Pro) versus *good* (Gemini 2.5 Flash Lite) candidate pairs, in the healthcare domain. Rows with significant differences (≥55% for one model) are highlighted: blue indicates *great* dominance, red indicates *good* dominance. Bold percentages show the dominant model.

| Refinement Type | Proportion | *Great* vs *Good* |
|---|---|---|
| Mandating explicit statements, justifications, or declarations | 16.7% | 52.6% vs 47.4% |
| Shifting focus from superficial to substantive qualities | 11.7% | 48.2% vs 51.8% |
| Adjusting scoring weights, granularity, or mechanisms | 8.9% | 48.5% vs 51.5% |
| Breaking down complex criteria into sub-components | 7.2% | **55.9%** vs 44.1% |
| Introducing penalties, prohibitions, or negative scoring | 6.6% | 43.5% vs **56.5%** |
| Replacing vague language with specific requirements | 6.2% | 54.7% vs 45.3% |
| Adding requirements for comparing alternatives | 5.9% | 49.0% vs 51.0% |
| Broadening criteria to accept multiple approaches | 5.6% | 44.4% vs **55.6%** |
| Adding conditional or context-dependent rules | 4.5% | 51.4% vs 48.6% |
| Streamlining by removing redundancy | 4.4% | 41.6% vs **58.4%** |
| Adding timing, sequencing, or process flow criteria | 3.5% | 54.1% vs 45.9% |
| Mandating precise language or technical accuracy | 3.3% | 50.2% vs 49.8% |
| Requiring causal explanations or mechanistic understanding | 3.3% | 51.8% vs 48.2% |
| Enhancing verification, validation, and evidence standards | 2.3% | **55.0%** vs 45.0% |
| Mandating specific structure or formatting | 2.1% | 53.8% vs 46.2% |
| Requiring explicit justification for decisions | 1.9% | 50.3% vs 49.7% |
| Defining explicit scope, boundaries, or constraints | 1.8% | **58.9%** vs 41.1% |
| Incorporating risk analysis or safety constraints | 1.8% | **55.2%** vs 44.8% |
| Requiring specific, actionable recommendations | 1.0% | **55.5%** vs 44.5% |
| Correcting errors or aligning with intended standards | 0.9% | 32.8% vs **67.2%** |
| Assessing communication quality or tone | 0.5% | 43.8% vs **56.2%** |

## 6 RELATED WORK

**Reward over-optimization.** Gao et al. (2023) highlighted the issue of reward over-optimization for both best-of-n sampling and reinforcement learning when using preference-based reward models. Although this phenomenon has since been repeatedly observed in empirical studies (Bai et al., 2022; Moskovitz et al., 2023; Perez et al., 2023; Gui et al., 2024; Wang et al., 2024), its theoretical underpinnings remain limited. Existing analyses typically relate the performance degradation caused by a proxy reward to global statistics describing how far the proxy deviates from the true reward (Huang et al., 2025a; Mroueh, 2024). In contrast, our work provides a sharper perspective: what truly governs performance is the fidelity of the proxy reward in the high-value region, where high-quality responses concentrate.

**Rubrics reward.** RL from rubrics reward (RLRR) has proven to be an effective method in specialized domains like science and health (Gunjal et al., 2025), general instruction-following (Huang et al., 2025b; Viswanathan et al., 2025), and for enhancing agentic ability (Team et al., 2025), with implementations using both online and offline RL. The idea of rubrics is also utilized in generative reward models (GRMs), wherein a reward model is prompted to first generate rubrics and then use them to evaluate a response (Liu et al., 2025b; Chen et al., 2025). This approach enables inference time scaling of reward modeling and improves explainability. However, generating rubrics on the fly is computationally inefficient and unsuitable for large-scale training.

## 7 DISCUSSION

In this paper, we investigate rubric-based reward modeling for LLM post-training. We begin by analyzing the central weakness of reinforcement fine-tuning, *reward over-optimization*, and theoretically trace it to misspecification of the proxy reward in the high-reward tail. A comprehensive empirical study highlights rubric-based rewards as an effective remedy. We further demonstrate that carefully designed rubrics, which distinguish among *great*, *diverse* off-policy responses, lead to consistently strong fine-tuning performance.

**Off-policy responses for Bradley-Terry reward model training might generalize, but is sample inefficient.** While we find a medium amount off-policy responses ($n = 5000$, in addition to the same number of on-policy responses) do not help Bradley-Terry reward model guide the current policy (see Appendix D), we note that other work successfully train BT reward model with off-policy samples, but with a much larger scale—using up to 20 million high quality samples ((Liu et al., 2025a; Cui et al., 2023)). Indeed, Bradley-Terry reward model's generalizability scales with the number, and diversity of training samples. However, it's not always easy to find large-scale data for many specialized domains, such as healthcare. In contrast, rubric-based reward can easily encode generalizable principles from limited amount of data.

**Weighted average of rubric score is not optimal.** To specifically analyze the impact of rubric quality, we deliberately use the most simple method of score aggregation, by taking a weighted average of scores from the satisfied criteria. Prior work has explored diverse approaches, including implicit aggregation by a verifier model (Gunjal et al., 2025), sophisticated frameworks to capture non-linear dependencies (Huang et al., 2025b), weighted averages of continuous scores (Viswanathan et al., 2025), and model-based self-critique that weighs criteria against internal priors (Team et al., 2025). We acknowledge that aggregation is a central component of an optimal rubric reward system and leave it for future work.

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

We fully understand the responsibility of using LLMs in academic research. We carefully monitor any potential problems, such as plagiarism or scientific misconduct (e.g., fabrication of facts) when using LLMs. We make sure these problems do not occur in the paper.

## A  THEORETICAL RESULTS

**Theorem 1.** *Suppose each $R_0^x \sim U(0,1)$ and $f(R_0^x) \stackrel{d}{=} R_0^x$. Then it holds that:*

*(i) KL divergence is invariant to $f$:*

$$\mathbb{D}_{\mathrm{KL}}\left[\pi_r(y \mid x)\|\pi_0(y \mid x)\right] = \frac{(1/\beta - 1)e^{1/\beta} + 1}{e^{1/\beta} - 1} - \log \beta - \log(e^{1/\beta} - 1).$$

*(ii) Expected reward (or win rate) of $\pi_r$ is $\frac{\int_0^1 f^{-1}(u)e^{u/\beta}\mathrm{d}u}{\beta(e^{1/\beta}-1)}$. [Proof].*

*Proof.* First, we compute the KL divergence. When $f(R_0^x) \sim U(0,1)$, by Proposition 1, the KL divergence is

$$\mathbb{D}_{\mathrm{KL}}\left[\pi_r(y \mid x)\|\pi_0(y \mid x)\right] = \mathbb{E}_{x \sim D}\left[\frac{\mathbb{E}\left[f(R_0^x)\,e^{f(R_0^x)/\beta}/\beta\right]}{\mathbb{E}\left[e^{f(R_0^x)/\beta}\right]} - \log \mathbb{E}\left[e^{f(R_0^x)/\beta}\right]\right]$$

$$= \mathbb{E}_{x \sim D}\left[\frac{\int_0^1 ue^{u/\beta}\mathrm{d}u}{\beta \int_0^1 e^{u/\beta}\mathrm{d}u} - \log\left(\int_0^1 e^{u/\beta}\mathrm{d}u\right)\right] = \frac{(1/\beta - 1)e^{1/\beta} + 1}{e^{1/\beta} - 1} - \log\left[\beta(e^{1/\beta} - 1)\right].$$

Then, we compute the expected reward: denote $T_0^x = f(R_0^x)$,

$$\mathbb{E}_{x \sim D,\, y \sim \pi_r(\cdot|x)}\left[r^\star(x,y)\right] = \mathbb{E}_{x \sim D}\left[\frac{\mathbb{E}\left[R_0^x\,e^{f(R_0^x)/\beta}\right]}{\mathbb{E}\left[e^{f(R_0^x)/\beta}\right]}\right]$$

$$= \mathbb{E}_{x \sim D}\left[\frac{\mathbb{E}\left[f^{-1}(T_0^x)\,e^{T_0^x/\beta}\right]}{\mathbb{E}\left[e^{T_0^x/\beta}\right]}\right] = \mathbb{E}_{x \sim D}\left[\frac{\int_0^1 f^{-1}(u)e^{u/\beta}\mathrm{d}u}{\int_0^1 e^{u/\beta}\mathrm{d}u}\right]$$

$$= \frac{\int_0^1 f^{-1}(u)e^{u/\beta}\mathrm{d}u}{\int_0^1 e^{u/\beta}\mathrm{d}u} = \frac{\int_0^1 f^{-1}(u)e^{u/\beta}\mathrm{d}u}{\beta(e^{u/\beta} - 1)}$$

Since $F_0^x(R_0^x) = R_0^x$ when $R_0^x \sim U(0,1)$, the win rate is the expected reward. Then the theorem follows.

$\square$

## B    PROMPTS USED FOR EXPERIMENTS

---

**Prompt for Constructing Initial Rubrics**

You're a skilled judge evaluating the quality of LLM responses to a user prompt. Your first task is to create a comprehensive rubric for grading these responses across multiple dimensions.

Given a user prompt, generate a list of binary (yes/no) criteria. These criteria should assess how well the LLM answered the prompt. Only write rubrics you are confident about.

Here are tips for writing good rubrics:

i. MECE:
    - Mutually Exclusive, Collectively Exhaustive

ii. Completeness:
    - Consider all the elements you would want to include to create a perfect response and put them into the rubric. This means including not only the facts and statements directly requested by the prompt, but also the supporting details that provide justification, reasoning, and logic for your response. Each of these elements should have a criterion because each criterion helps to develop the answer to the question from a slightly different angle.

iii. No overlapping:
    - the same error from a model shouldn't be punished multiple times.

iv. Diversity:
    - The rubric items should include variable types of information.
    - If all criteria are like "the response mentions A", "the response mentions B", then this is not a good rubric.

v: How many rubric items for each prompt
    - There is no golden standard, and the desired number of rubrics varies by accounts and task types.
    - Write rubrics that cover all aspects of an ideal response.

vi: How many rubric items to fail
    - A good rule of thumb is that the model fails on 50% of rubrics items

vii: Atomicity / Non-stacked
    - Each rubric criterion should evaluate exactly one distinct aspect. Avoid bundling multiple criteria into a single rubric. Most stacked criteria with the word "and" can be broken up into multiple pieces.
    ✘ Response identifies George Washington as the first U.S. president and mentions he served two terms.
    ✔ Response identifies George Washington as the first U.S. president.
    ✔ Response mentions that George Washington served two terms.

viii: Specificity
    - Criteria should be binary (true or false) and objective.
    - Avoid vague descriptions (e.g., "the response must be accurate" is vague).
    - Example: "The response should list exactly three examples."

---

ix: Self-contained
- Each criterion should contain all the information needed to evaluate a response, e.g.
✘ Mentions the capital city of Canada.
✔ Mentions the capital city of Canada is Ottawa.

x: Criterion should be verifiable without requiring external search.
✘ Response names any of the Nobel Prize winners in Physics in 2023
✔ Response names any of the following Nobel Prize winners in Physics in 2023: Pierre Agostini, Ferenc Krausz, or Anne L'Huillier.

xi. The binary criteria should be phrased so that yes means the model response is good and no means the model response is bad.

Finally, we want to assign different weight for each question. Give a weight on a scale of 1 (least important) to 3 (most important) for each question based on
1. the question's alignment with user demand (3 if user would be frustrated if the answer is no; 1 if user would not be bothered at all if the answer is no)
2. the question's importance in terms of determining quality/correctness (3 if the response would be completely incorrect if the answer is no; 1 if an extreme edge case would be missed and the overall quality won't be affected if the answer is no)

Here is the user prompt for which we want to generate a rubric:

**PROMPT:**
{prompt}

Return ONLY the JSON array of the rubrics, no other text. For example:

```
[
  {{"criterion": "Does the response provide a list of
     songs?", "weight": 3}},
  {{"criterion": "The response explicitly state it is
     listing French romantic songs.", "weight": 2}}
]
```

Note: Local IDs will be automatically assigned to each criterion (c1, c2, c3, etc.), so don't include IDs into outputed criterion.

---

**Prompt for Improving Rubrics**

You're a skilled judge assessing the quality of LLM responses to a user prompt. The current rubric isn't good enough to effectively differentiate between high-quality responses.

Your goal is to improve the current rurbics to address this (adding new creteria, rewriting, decomposing, and deleting the current creteria). The updated rubric must be comprehensive and consistently applicable for grading LLM responses. These criteria should specifically assess how well the LLM answered the given prompt. Only write rubrics you are confident about.

Here are tips for writing good rubrics:

i. MECE:
- Mutually Exclusive, Collectively Exhaustive

ii. Completeness:
- Consider all the elements you would want to include to create a perfect response and put them into the rubric. This means including not only the facts and statements directly requested by the prompt, but also the supporting details that provide justification, reasoning, and logic for your response. Each of these elements should have a criterion because each criterion helps to develop the answer to the question from a slightly different angle.

iii. No overlapping:
- the same error from a model shouldn't be punished multiple times.

iv. Diversity:
- The rubric items should include variable types of information.
- If all criteria are like "the response mentions A", "the response mentions B", then this is not a good rubric.

v: How many rubric items for each prompt
- There is no golden standard, and the desired number of rubrics varies by accounts and task types.
- Write rubrics that cover all aspects of an ideal response.

vi: How many rubric items to fail
- A good rule of thumb is that the model fails on 50% of rubrics items

vii: Atomicity / Non-stacked
- Each rubric criterion should evaluate exactly one distinct aspect. Avoid bundling multiple criteria into a single rubric. Most stacked criteria with the word "and" can be broken up into multiple pieces.
✘ Response identifies George Washington as the first U.S. president and mentions he served two terms.
✔ Response identifies George Washington as the first U.S. president.
✔ Response mentions that George Washington served two terms.

viii: Specificity
- Criteria should be binary (true or false) and objective.
- Avoid vague descriptions (e.g., "the response must be accurate" is vague).
- Example: "The response should list exactly three examples."

ix: Self-contained
- Each criterion should contain all the information needed to evaluate a response, e.g.
✘ Mentions the capital city of Canada.
✔ Mentions the capital city of Canada is Ottawa.

x: Criterion should be verifiable without requiring external search.
✘ Response names any of the Nobel Prize winners in Physics in 2023
✔ Response names any of the following Nobel Prize winners in Physics in 2023: Pierre Agostini, Ferenc Krausz, or Anne L'Huillier.

xi. The binary criteria should be phrased so that yes means the model response is good and no means the model response is bad.

Finally, we want to assign different weight for each criterion. Give a weight on a scale of 1 (least important) to 3 (most important) for each question based on
1. the question's alignment with user demand (3 if user would be frustrated if the answer is no; 1 if user would not be bothered at all if the answer is no)
2. the question's importance in terms of determining quality/correctness (3 if the response would be completely incorrect if the answer is no; 1 if an extreme edge case would be missed and the overall quality won't be affected if the answer is no)

Here is the user prompt for which we want to improve the rubric:

**PROMPT:**
{prompt}

The existing rubrics we are using is:
{rubrics}

The two reference responses are:

**Reponse 1:**
{response1}

**Reponse 2:**
{response2}

Return ONLY the JSON array of the full rubrics, no other text. For example:

```
[
  {{"criterion": "Does the response provide specific
     release years for each song?", "weight": 2}},
  {{"criterion": "The response includes artist names
     for each song mentioned", "weight": 1}}
]
```

Note: Local IDs will be automatically assigned to each criterion, so don't include IDs in your output.

---

**Prompt for Scoring Responses**

You are a skilled judge who will be assessing the quality of LLM responses to a user prompt.

Given a user prompt, LLM response, and a rubric, your task is evaluating the performance of the model response by seeing whether or not it meets the rubric dimension.

Answer the each of the given rubric dimension in either "yes" or "no". Do not output any response other than "yes" or "no".

Keep in mind that you will be grading industry-leading LLMs. Make sure to have high expectation for grading the responses.

Make sure your evaluation is as objective and consistent as it could be. By consistent we mean that a different evaluator's assessment of the task should agree with yours.

Think carefully before you make the decision. After you make the decision, explicitly output which dimension receives "yes" and which dimension receives "no".

**Input:**

* **PROMPT:** {prompt}

* **RESPONSE:** {response}

* **RUBRIC:** {rubric}

Return ONLY the JSON array, no other text. For example:

```
{{"c1":"yes", "c2":"no", "c3":"yes"}}
```

## C Hyperparameter

The GRPO hyperparameters used for the RLRR results in Table 1 are listed in Table 4. The finance domain is an exception due to the limited amount of available data, and its hyperparameters are presented separately in Table 5. Figure 4 uses a longer training run than in table 1, resulting in some performance variations due to different warmup steps.

Table 4: GRPO Hyperparameter Configuration

| Hyperparameter | Value |
| --- | --- |
| Rollouts per Prompt | 16 |
| Gradient Accumulation Steps | 2 |
| Per-Device Train Batch Size | 6 |
| Warmup Ratio | 0.1 |
| KL Coefficient | 0.01 |
| Learning Rate | $1.0 \times 10^{-5}$ |
| Learning Rate Scheduler | Constant with Warmup |
| Maximum Sequence Length | 3584 |
| Training Epochs | 2 |

Table 5: GRPO Hyperparameter Configuration in Finance Domain

| Hyperparameter | Value |
| --- | --- |
| Rollouts per Prompt | 16 |
| Gradient Accumulation Steps | 2 |
| Per-Device Train Batch Size | 6 |
| Warmup Ratio | 0.1 |
| KL Coefficient | 0.01 |
| Learning Rate | $1.0 \times 10^{-5}$ |
| Learning Rate Scheduler | Constant with Warmup |
| Maximum Sequence Length | 3584 |
| Training Steps | 320 |

## D Empirical Results on RLHF

We finetune a Bradley-Terry Reward model on various responses, with preference generated by GPT-4.1, the same model as the judge model for evaluation. For each of the prompt in the training set, we generated a pair of responses at temperature 1.0 using the base policy model Qwen3-8B-Base (on-policy) or Gemini-2.5-Pro (off-policy). Preferences were labeled using GPT-4.1, the same model used for final evaluation. This preference data was then used to train a reward model based on Llama-3.1-8B-Instruct, with hyperparameters specified in Table 7. Finally, this reward model was used for GRPO training, following the configuration in Table 4.

We find that using on-policy responses is a baseline that can't be easily improved upon:

1. Training on off-policy, *great* responses deteriorates the performance

2. Adding both off-policy and on-policy responses only helps with win rates but not helps with healthbench. This suggests that the off-policy samples only help the reward model encode superficial features (that can game LLM-judge) instead of true capabilities as measured by more objective metrics.

This experiment shows the difficulty of improving Bradley-Terry models with off-policy responses.

Table 6: Win-rates and HealthBench scores for the Health domain.

| Method | Win-Rate | HealthBench |
|---|---|---|
| Reward Model (on-policy) | 26.8% | 0.3036 |
| Reward Model (off-policy, *great*) | 22.4% | 0.2798 |
| Reward Model (on-policy + off-policy) | 30.7% | 0.3032 |
| SFT | 25.8% | 0.3094 |
| Initial, Prompt only | 21.7% | 0.3004 |
| 1 *good* Pair | 22.4% | 0.2912 |
| 1 *great* Pair | **26.5%** | **0.3163** |
| 4 *great* Pairs | 31.4% | 0.3348 |
| 4 *great* & Diverse Pairs | **34.4%** | **0.3513** |

Table 7: Reward Model Hyperparameter Configuration

| Hyperparameter | Value |
|---|---|
| Learning Rate | $1.0 \times 10^{-5}$ |
| Per-Device Train Batch Size | 4 |
| Gradient Accumulation Steps | 4 |
| Training Epochs | 10 |
| Maximum Sequence Length | 8192 |
| Warmup Ratio | 0.1 |
| Learning Rate Scheduler | Cosine |

## E  LLM JUDGE FOR EVALUATION

We use the same judge model as the rubrics proposer (GPT-4.1). This is by design: our primary goal is to test how best to incorporate additional responses into the rubric construction process. By using the same powerful model for both proposing rubrics and evaluating final outputs, we isolate the quality of the candidate responses as the key experimental variable and eliminate potential confounding issues that could arise from disagreements between a proposer and a judge.

We use a minimal judge prompt to compare two responses:

---

**LLM Judge Prompt**

You are a skilled judge who will be assessing the quality of LLM responses to a user prompt.

Avoid any position biases and ensure that the order in which the responses were presented does not influence your decision. Do not allow the length of the responses to influence your evaluation.

Here is the user prompt:

PROMPT: {prompt}

The two responses are:

Response 1: {response1}

Response 2: {response2}

Which reponse would you prefer? Enclose your final answer (1 or 2) in `\boxed{{...}}`.

---

To reduce the position bias, we randomly flipped two responses.

## F    FRONTIER MODELS USED TO CREATE CANDIDATE RESPONSES

The 16 frontier models used to generate candidate responses are:

- Gemini-2.5-Pro
- Gemini-2.5-Flash
- GPT-5
- GPT-4.1
- GPT-4o-2024-05-13
- o3
- o1-2024-12-17
- o4-mini
- Claude-Sonnet-4-20250514
- Claude-3-7-Sonnet-20250219
- Deepseek-V3
- Deepseek-R1
- Kimi-K2-Instruct
- GLM-4.5
- Qwen3-235B-A22B-Instruct-2507
- Mistral-Medium-Latest

## G    PRINCIPLES OF SELECTING PROMPTS

We manually curated a prompt dataset according to a specific set of criteria to ensure quality and suitability for rubrics training.

- Prompts have clear user intent.
- Prompts are not multimodal/search/trivia/GTFA.
- Prompts are not too simple.
- Prompts are generalist friendly (nothing technical that requires expert knowledge).
- Prompts are not open-ended / creative-writing.
- Prompts are designed such that there is an objectively better response ("Tell me a good bedtime story" can have multiple good responses)

## H    PATTERN DETECTION ON RUBRIC REFINEMENTS

In Table 3 we show clusters of various rubric-refinements. To get this result, we first prompt GPT-4.1 to describe the differences between refined rubrics, and the initial rubrics (for refinement with one-*good*-pair, and one-*great*-pair). The differences are listed as atomic items describing different aspects of improvement. We then run a clustering algorithm on the atomic differences from both one-*good*-pair, and one-*great*-pair (they have almost the same number of atomic differences) to get interpretable clusters, and report the proportions from one-*good*-pair versus and one-*great*-pair

## I  EXAMPLES OF RUBRICS AND RUBRIC REFINEMENTS

In this example, the initial rubric cannot differentiate between the two *great* responses (both from Gemini-2.5-Pro). This is because it primarily handles basic criteria (e.g. the diagnosis is correct), which both responses satisfy. The refinement process resolves the tie by adding a new criterion **c7**—requiring the mention of an imaging test to confirm the diagnosis. Such refinement is one example for the cluster **Enhancing verification, validation, and evidence standards** identified in Table 3.

---

### Rubric Refinement Example

**Prompt**
*What is the likely diagnosis for a young girl with a history of repeated pain over the medial canthus and chronic use of decongestants, who now presents with intense chills, rigors, diplopia on lateral gaze, and a congested optic disc on examination?*

---

**Response 1**
Most likely diagnosis: **Cavernous sinus thrombosis (CST)**. This is a life-threatening medical emergency.

*Why CST fits*

- Medial canthus pain + decongestants $\rightarrow$ sinusitis (ethmoid/sphenoid)/dacryocystitis; valveless ophthalmic veins communicate with cavernous sinus.
- Intense chills/rigors $\rightarrow$ septicemia from septic thrombophlebitis.
- Diplopia on lateral gaze $\rightarrow$ abducens nerve (CN VI) involvement.
- Congested optic disc $\rightarrow$ impaired venous outflow / increased ICP.

*Differential:* orbital cellulitis; meningitis/brain abscess.
*Immediate management:* urgent contrast CT or MRI/MRV; high-dose IV antibiotics; consider anticoagulation; drain source if indicated.

**Response 2**
Most likely diagnosis: **Cavernous sinus thrombosis (CST)**, secondary to dacryocystitis/orbital cellulitis. This is a life-threatening emergency.

*Clinical reasoning*

- Medial canthus pain localizes to lacrimal sac; chronic congestion/decongestants imply naso-lacrimal dysfunction and recurrent dacryocystitis.
- Valveless facial/ophthalmic veins permit retrograde spread to cavernous sinus.

*CST manifestations*

- Sepsis (chills, rigors).
- CN VI palsy causing diplopia on lateral gaze.
- Papilledema from impaired venous drainage / increased ICP.

*Urgency & treatment* medical/neurosurgical emergency; high-dose IV antibiotics essential.

---

**Rubric Scores (Initial)**

| Criterion (Initial) | Weight | R1 (0/1) | R2 (0/1) |
| --- | --- | --- | --- |
| c1: Identifies CST as most likely diagnosis | 3 | 1 | 1 |
| c2: States it is a medical emergency | 3 | 1 | 1 |
| c3: Links medial canthus pain + decongestants to sinusitis | 3 | 1 | 1 |
| c4: Diplopia due to CN VI involvement | 3 | 1 | 1 |
| c5: Papilledema from impaired venous drainage/ICP | 2 | 1 | 1 |
| c6: Chills/rigors = systemic infection/bacteremia | 2 | 1 | 1 |
| c7: Includes medical disclaimer / seek care | 2 | 0 | 0 |
| c8: Mentions orbital cellulitis differential | 1 | 1 | 1 |
| c9: Mentions high-dose IV antibiotics | 1 | 1 | 1 |

*Weighted total (Initial):* R1 $= 18/20$, R2 $= 18/20$

---

**Rubric Scores (Refined)**

| Criterion (Refined) | Weight | R1 (0/1) | R2 (0/1) |
| --- | --- | --- | --- |
| c1: Identifies CST as most likely diagnosis | 3 | 1 | 1 |
| c2: Explicitly states CST is a medical emergency | 3 | 1 | 1 |
| c3: Links medial canthus pain + decongestants to sinusitis/dacryocystitis | 3 | 1 | 1 |
| c4: Diplopia due to abducens (CN VI) involvement | 3 | 1 | 1 |
| c5: Papilledema from impaired venous drainage/ICP | 2 | 1 | 1 |
| c6: Sepsis secondary to CST (chills/rigors) | 2 | 1 | 1 |
| **c7: Urgent imaging (contrast CT or MRI/MRV) required to confirm diagnosis** | 2 | 1 | 0 |
| c8: High-dose IV antibiotics are initial mainstay | 2 | 1 | 1 |
| c9: Medical disclaimer / seek immediate care | 2 | 0 | 0 |
| c10: Mentions orbital cellulitis differential | 1 | 1 | 1 |
| c11: Notes other CNs (III, IV, V1, V2) may be affected | 1 | 1 | 0 |
| c12: Avoids incorrect primary diagnosis | 3 | 1 | 1 |

*Weighted total (Refined):* R1 $= 25/27$, R2 $= 22/27$

## J   RUBRICS CATEGORIZATION

We find the constructed rubrics encode diverse and concrete criteria instead of superficial stylistic features. To see this, we systematically study rubrics for the health domain constructed with the workflow using 4 pairs from *Great & Diverse* candidate responses. Since each rubric criterion assesses certain capability of policy models, we apply hierarchical K-means clustering to the targeted capabilities of each criterion. This clustering analysis yields 20 types of capabilities, with their distribution presented in Figure 5. For each type, we illustrate its meaning with an example in the following box. We also provide the full clustering results with all examples in the supplement material.

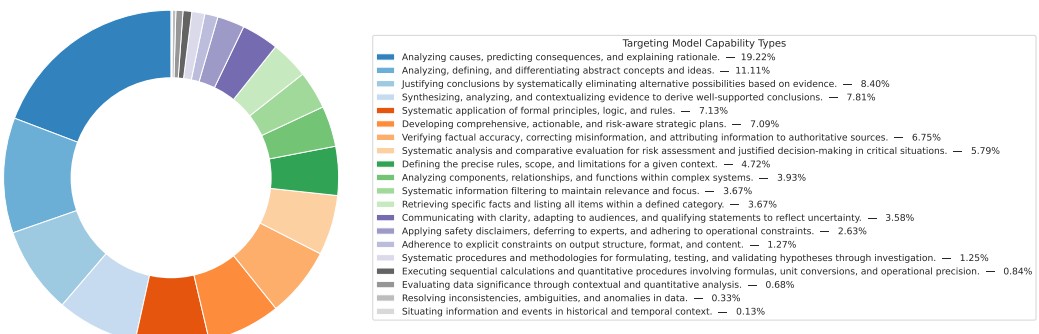

Figure 5: The Distribution of Rubrics Targeting Each Type of Model Capabilities

---

**Categorization of Target Capabilities of Rubrics**

**Target Capability Type 1**
Analyzing causes, predicting consequences, and explaining rationale

**Example:**
**Prompt:** A 62-year-old man presents with an inability to tan and mild fatigue. Laboratory tests indicate mild anemia with hemoglobin at 11 g/dL, decreased haptoglobin, and 3% reticulocytes. Given these findings and the peripheral blood smear results suggesting hemolytic anemia, what specific patient history or characteristic is most consistent with the mechanical destruction of red blood cells as the underlying cause?

**Rubric:** The response explains that decreased haptoglobin is due to its binding to free hemoglobin released during red blood cell lysis.

**Target Model Capability:** Connecting specific observations or data points to their corresponding causes, effects, or implications within a larger system.

**Target Capability Type 2**
Systematic analysis and comparative evaluation for risk assessment and justified decision-making in critical situations.

**Example:**
**Prompt:** A patient presents in a stuporous condition, with a history of persistent sadness, suicidal attempts, and a lack of eating and sleeping according to the patient's parents. What is the recommended treatment in this case?

**Rubric:** The response recommends inpatient hospitalization for safety, evaluation, and stabilization (not outpatient or home-based care).

**Target Model Capability:** Identifying critical conditions requiring immediate and decisive intervention based on a severity assessment.

**Target Capability Type 3:**
Developing comprehensive, actionable, and risk-aware strategic plans.

**Example:**
**Prompt:** What is the ideal management for a 5-year-old boy with retinoblastoma involving the entire right eyeball and 2-3 small lesions in the periphery of the other eye?

**Rubric:** The response specifies the timing of prosthetic eye fitting after enucleation (e.g., 4-6 weeks post-surgery).

**Target Capability:** Detailing a post-intervention protocol by specifying follow-on diagnostic assessments and subsequent restorative procedures with timelines.

**Target Capability Type 4:**
Structured and logically clear presentation.

**Example:**
**Prompt:** What is the most likely diagnosis for a 3-year-old child who presents with eczematous dermatitis on extensor surfaces and has a mother with a history of bronchial asthma?

**Rubric:** The response is organized logically, with clear separation between diagnosis, reasoning, differential diagnoses, and any disclaimers.

**Target Capability:** Structuring the response logically by partitioning distinct conceptual elements into clearly delineated sections.

**Target Capability Type 5:**
Analyzing components, relationships, and functions within complex systems.

**Example:**
**Prompt:** What is the most probable diagnosis for a 6-year-old boy who has been experiencing headaches and peripheral vision loss for four months, with a CT scan showing a suprasellar mass with calcification?

**Rubric:** The response links peripheral vision loss to compression of the optic chiasm by the mass.

**Target Capability:** Justifying a conclusion by explicitly linking individual pieces of evidence to their respective supporting roles in the final determination.

**Target Capability Type 6:**
Executing sequential calculations and quantitative procedures involving formulas, unit conversions, and operational precision.

**Example:**
**Prompt:** A 1-year-old child weighing 6 kg is suffering from Acute Gastroenteritis, showing signs of sunken eyes and a skin pinch test indicating rapid fluid replenishment. Based on these symptoms, what volume and rate of Ringer's Lactate infusion would you administer over the first six hours?

**Rubric:** The response provides the correct infusion rates for each phase: 180 mL/hour for the first hour and 84 mL/hour for the next five hours.

**Target Capability:** Executing a precise, multi-step quantitative calculation based on given inputs and established formulas to derive a phased implementation plan.

**Target Capability Type 7:**
Resolving inconsistencies, ambiguities, and anomalies in data.

**Example:**
**Prompt:** A patient with a head injury is admitted to the intensive care unit showing signs of raised intracranial pressure. He is placed on a ventilator and given intravenous fluids and diuretics. After 24 hours, the patient's urine output is 3.5 liters, serum sodium level is 156

mEq/l, and urine osmolality is 316 mOsm/kg. What is the most likely cause of these clinical findings?

**Rubric:** The response identifies the urine osmolality of 316 mOsm/kg as inappropriately low for the degree of hypernatremia (i.e., urine should be more concentrated in this context).

**Target Capability:** Evaluating the relationship between multiple variables to identify paradoxical or inconsistent patterns relative to expected system behavior.

**Target Capability Type 8:**
Verifying factual accuracy, correcting misinformation, and attributing information to authoritative sources.

**Example:**
**Prompt:** According to the latest resuscitation guidelines, for how long must umbilical cord clamping be delayed in preterm infants?

**Rubric:** The response identifies at least one authoritative organization issuing the guideline (e.g., AHA, ILCOR, ACOG, WHO, ERC, NRP).

**Target Capability:** Attribute the factual knowledge to the authoritative sources.

**Target Capability Type 9:**
Synthesizing, analyzing, and contextualizing evidence to derive well-supported conclusions.

**Example:**
**Prompt:** A 33-year-old woman is brought to the emergency department 15 minutes after being stabbed in the chest with a screwdriver. Given her vital signs of pulse 110/min, respirations 22/min, and blood pressure 90/65 mm Hg, along with the presence of a 5-cm deep stab wound at the upper border of the 8th rib in the left midaxillary line, which anatomical structure in her chest is most likely to be injured?

**Rubric:** The response provides a clear, logical synthesis connecting wound location, anatomical relationships, and clinical findings to justify the conclusion.

**Target Capability:** Synthesizing multiple distinct lines of evidence into a coherent, logical argument to justify a final conclusion.

**Target Capability Type 10:**
Analyzing, defining, and differentiating abstract concepts and ideas.

**Example:**
**Prompt:** A 68-year-old woman with elevated serum calcium, high parathyroid hormone, low phosphorus, and a history of kidney stones presents with fatigue, constipation, diffuse bone pain, and a 24-hour urine calcium level that is elevated. Given these clinical and laboratory findings, what radiologic finding on a hand X-ray would confirm the suspected diagnosis of this patient's condition?

**Rubric:** The response accurately distinguishes between primary and secondary hyperparathyroidism if mentioned.

**Target Capability:** Differentiating between closely related sub-categories of a primary concept based on their defining features.

**Target Capability Type 11:**
Defining the precise rules, scope, and limitations for a given context.

**Example:**
**Prompt:** You are called to evaluate a newborn who was born yesterday to a 39-year-old mother. Upon examination, what chromosomal abnormality is most likely responsible for the observations typically associated with Down syndrome?

**Rubric:** The response recommends or references karyotype analysis or equivalent genetic testing as the definitive diagnostic method for confirming the chromosomal abnormality.

**Target Capability:** Specifying the definitive method or standard procedure required for confirmation or validation.

**Target Capability Type 12:**
Communicating with clarity, adapting to audiences, and qualifying statements to reflect uncertainty.

**Example:**
**Prompt:** Which complication during pregnancy is least likely to increase the risk of postpartum uterine atonicity and why?

**Rubric:** The response uses cautious and appropriate phrasing (e.g., 'least likely,' 'low association') rather than making absolute claims of zero risk.

**Target Capability:** Calibrating language precisely to reflect nuances and uncertainty, avoiding absolute or overly definitive statements.

**Target Capability Type 13:**
Applying safety disclaimers, deferring to experts, and adhering to operational constraints.

**Example:**
**Prompt:** Considering the patient's history and current presentation of sudden right arm weakness, numbness, facial drooping, and slurred speech, what is the strongest predisposing factor contributing to his condition?

**Rubric:** The response avoids providing direct medical advice and, if appropriate, includes a disclaimer to seek immediate professional medical attention for stroke symptoms.

**Target Capability:** Adhering to predefined safety protocols or operational constraints by including appropriate disclaimers.

**Target Capability Type 14:**
Evaluating data significance through contextual and quantitative analysis.

**Example:**
**Prompt:** A 6-year-old boy presents with headache, cough, runny nose, and low-grade fever after being treated for a urinary tract infection with trimethoprim-sulfamethoxazole. He has a leukocyte count of 2,700/mm3 with a differential predominantly showing lymphocytes. What is the most likely underlying cause of his current symptoms?

**Rubric:** The response correctly interprets a leukocyte count of 2,700/mm3 as leukopenia for a 6-year-old child (normal range 5,000/mm3 - 15,000/mm3).

**Target Capability:** Accurately interpreting a quantitative data point by comparing it against a reference range to determine its significance.

**Target Capability Type 15:**
Systematic information filtering to maintain relevance and focus.

**Example:**
**Prompt:** A labourer involved with repair work of sewers presents with fever, jaundice, and renal failure. What is the most appropriate test to diagnose the suspected infection in this patient?

**Rubric:** The response is concise and focused on the diagnostic aspect, without excessive unrelated clinical management details.

**Target Capability:** Adhering strictly to the defined scope of a problem by excluding extraneous or irrelevant information.

**Target Capability Type 16:**
Systematic procedures and methodologies for formulating, testing, and validating hypotheses through investigation.

**Example:**
**Prompt:** Given an X-ray of a young man that shows heterotopic calcification around bilateral knee joints, what would be the next investigation to help diagnose the underlying condition?

**Rubric:** The response recommends creatine kinase (CK) testing if myositis or muscle involvement is considered in the differential.

**Target Capability:** Proposing specific, targeted investigative actions to differentiate between hypotheses or gather further evidence.

**Target Capability Type 17:**
Retrieving specific facts.

**Example:**
**Prompt:** Which nerves are associated with difficulty swallowing despite normal musculature function, and should be tested for their functionality?

**Rubric:** The response identifies the Facial nerve (Cranial Nerve VII) as relevant to the oral phase of swallowing (e.g., facial muscles, buccinator, taste, or saliva production).

**Target Capability:** Selecting specific entities from its internal knowledge that directly satisfy a given set of complex conditions.

**Target Capability Type 18:**
Situating information and events in historical and temporal context.

**Example:**
**Prompt:** A 10-year-old patient presents with tingling and numbness in the ulnar side of the finger. Four years ago, the patient sustained an elbow injury. Based on the symptoms and history, identify the fracture site that most likely occurred at the time of the initial accident.

**Rubric:** The response explicitly states or clearly implies that the patient's symptoms are a delayed complication (i.e., tardy onset) following the initial elbow injury.

**Target Capability:** Identifying and explicitly stating the temporal relationship between a past event and a current observation.

**Target Capability Type 19:**
Adherence to explicit constraints on output structure, format, and content.

**Example:**
**Prompt:** A 29-year-old pregnant woman at 10 weeks' gestation is experiencing progressively worsening nausea and vomiting, leading to a significant weight loss and affecting her ability to work. Despite taking ginger and vitamin B6, her symptoms persist. Her blood gas analysis indicates a pH of 7.43, pCO2 of 54 mmHg, and HCO3- of 31 mEq/L. What pharmacological intervention should be added to her treatment regimen to alleviate her symptoms?

**Rubric:** The response provides a clear, direct, and unambiguous recommendation for the next pharmacological agent to add.

**Target Capability:** Formulating a direct and unambiguous conclusion or recommendation that resolves the primary question.

**Target Capability Type 20:**
Justifying conclusions by systematically eliminating alternative possibilities based on evidence.

**Example:**
**Prompt:** What is the best intervention for hearing rehabilitation in a patient who has undergone surgery for bilateral acoustic neuroma?

**Rubric:** The response states that conventional hearing aids and CROS/BiCROS devices are

not effective for profound bilateral sensorineural hearing loss due to bilateral cochlear nerve loss.

**Target Capability:** Invalidating alternative solutions by providing a causal explanation for their ineffectiveness under given constraints.

## K DISTRIBUTION OF MODEL SOURCES IN REFINEMENT THROUGH DIFFERENTIATION

In the refinement with four diverse and great pairs, we gather responses from a set of SOTA models and adaptively select the best pair for RTD. We present the distribution of model sources for the responses used in the final refinement round in Figure 6. The responses are drawn from a diverse collection of models, with even the top model, Gemini-2.5-Pro, contributing only 13.64% of the responses

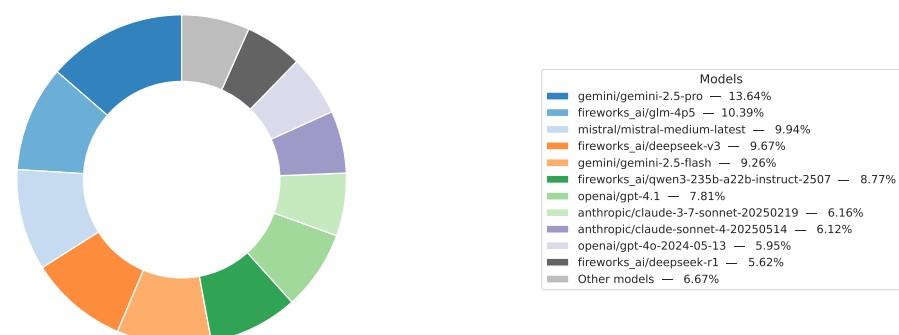

Figure 6: The Distribution of Model Sources in The Final Refinement Round

