# OpenReview forum: "Chasing the Tail: Effective Rubric-based Reward Modeling for Large Language Model Post-Training"
_ICLR.cc/2026/Conference — ICLR 2026 Poster_

### Official Review · Reviewer_ULR5 · 2025-10-22

**Soundness:** 3
**Presentation:** 2
**Contribution:** 3
**Rating:** 6
**Confidence:** 4

**Summary:**

This paper addresses the problem of reward over-optimization in the reinforcement fine-tuning (RFT) of Large Language Models (LLMs). This issue occurs when a policy model learns to achieve high scores from a proxy reward model without actually producing high-quality output, essentially "hacking" the reward. To address the issue, the authors propose a rubric-based reward refinement that distinguishes excellent responses from good ones, and distinguishes among a diverse set of off-policy responses.

**Strengths:**

The paper is well-motivated to address the reward hacking issue by designing a rubric-based reward system. The experiments are conducted across different settings (1/4 pairs, good v.s. great v.s. great and diverse pairs). There are also analyses on refinement types to give more detailed examination of the proposed method.

**Weaknesses:**

The technical components—LLMs as proposers/verifiers and RFT—are not very novel

The paper's presentation could be improved. The initial motivation in Section 3 is well-reasoned but contains excessive detail better suited for the appendix. In contrast, Section 5.3 would benefit from more in-depth analysis of refinement types and their specific details.

**Questions:**

1. I'm curious if there are ways to attribute the language model performance gain to specific refinement types. For example, certain refinements, such as "Defining explicit scope, boundaries, or constraints" mentioned in Table 3 might be particularly useful and could be explicitly incorporated into the rubric-writing instructions.


2. Writing rubrics and generating evaluations can also be regarded as actions and optimized through reinforcement learning. With a rubric-writing LLM-as-judge, we could train it jointly with the base LLM. This could create an interesting multi-agent learning paradigm to co-evolve the rubric-writer and underlying LLMs. Any thoughts on such extensions?

---

> ### Author Response · Authors · 2025-11-23
> **Response to Reviewer ULR5**
>
> ## Novelty
> > **W1**: The technical components—LLMs as proposers/verifiers and RFT—are not very novel
>
> **Our response**:
>
> We agree this LLMs-as-proposed/verifiers-and-RFT paradigm exists for a long time. Our contribution does not lie in claiming novelty for this paradigm itself. Rather, our focus is on studying how to leverage off-policy responses within this framework to mitigate reward hacking. We adopt this paradigm because it offers clear practical advantages for our investigation, not because it is new.
>
> ## More details about refinement
>
> > **W2**: The paper's presentation could be improved. The initial motivation in Section 3 is well-reasoned but contains excessive detail better suited for the appendix. In contrast, Section 5.3 would benefit from more in-depth analysis of refinement types and their specific details.
> >
> > **Q1**: I'm curious if there are ways to attribute the language model performance gain to specific refinement types. For example, certain refinements, such as "Defining explicit scope, boundaries, or constraints" mentioned in Table 3 might be particularly useful and could be explicitly incorporated into the rubric-writing instructions.
>
> **Our response**:
>
> Thanks for the helpful suggestion. We will move more theoretical details in sec3 to the appendix in the final version once accepted. We’ve added more analyses of both rubrics themselves and refinements (during rubric construction) to explicitly show their reasonableness. Concrete examples of rubrics and refinements are provided in Appendix I and Table 3 (this is also in the previous version). We’ve further included category summaries for both rubrics and refinements in Appendices J and K (https://osf.io/7gt28/files/kaqxn?view_only=0e2cc586ceab4196bfcc5c4e0ac028a2 and https://osf.io/7gt28/files/5zsdg?view_only=0e2cc586ceab4196bfcc5c4e0ac028a2). For refinements, we’ve summarized some top refinement types based on average improvement of fine-tuned models by refined rubrics from the fine-tuned models by initial rubrics . Among the refinement types, we find that ``requirements for comparing alternatives``, ``explicit justifications for decisions``, and ``causal explanations`` are some of the more important refinements.
>
> We acknowledge this is preliminary correlation analysis. To attribute model performance increase to specific types of rubric refinements, require systematic causal interventions on the rubric-constructions, as well as large scale ablation experiments. This will be one very promising future direction. Thanks for pointing it out!
>
> ## Co-evolving rubric writer
>
> > **Q2**: Writing rubrics and generating evaluations can also be regarded as actions and optimized through reinforcement learning. With a rubric-writing LLM-as-judge, we could train it jointly with the base LLM. This could create an interesting multi-agent learning paradigm to co-evolve the rubric-writer and underlying LLMs. Any thoughts on such extensions?
>
> **Our response**:
> We appreciate this insightful suggestion. We think this co-evolving system is a promising future direction. One related work by Deepseek [1] demonstrates the potential of jointly training multiple components. While this lies beyond the scope of the current paper, we agree it represents an exciting avenue for future research.
>
> [1] Inference-Time Scaling for Generalist Reward Modeling

---

> > ### Comment · Reviewer_ULR5 · 2025-11-26
> > **Response by Reviewer**
> >
> > Thanks for the additional clarification. As noted by other reviewers, the reliability of using LLM as a judge and a rubric writer might be challenged, even if the tested domain works well with GPT4.1. Overall, I still appreciate the analysis and findings presented in the paper. Thus, I will maintain my assessment and keep the scoring.

---

> > > ### Author Response · Authors · 2025-11-26
> > >
> > > Thanks for your support and positive feedback. Besides using GPT-4.1 as an LLM judge, we also evaluate our rubric construction strategies on objective professional benchmarks (HealthBench and PrBench-Finance), and we present the full training dynamics at https://osf.io/7gt28/files/rzmjx?view_only=0e2cc586ceab4196bfcc5c4e0ac028a2. The results clearly support the benefits of leveraging high-quality and diverse responses in rubric construction.

---

### Official Review · Reviewer_Zt7D · 2025-11-02

**Soundness:** 3
**Presentation:** 3
**Contribution:** 2
**Rating:** 4
**Confidence:** 4

**Summary:**

This paper proposes an iterative rubric refinement framework (“Refinement-through-Differentiation”, RTD) to construct reward models that focus on the high-reward tail of LLM responses, aiming to mitigate reward over-optimization in reinforcement fine-tuning (RFT). The authors provide a theoretical justification that reward model accuracy in the top quantile of responses is the dominant factor for alignment success, and empirically show that rubric-based rewards—refined using high-quality off-policy responses—outperform baselines in both general and medical domains.

While the motivation is compelling and the empirical results appear promising, I have significant concerns regarding bias propagation, response diversity vs. superficial style differences, and the robustness of the “pair” construction strategy. These issues undermine confidence in whether the observed gains truly reflect improved capability discrimination rather than alignment to a specific LLM’s stylistic or judgmental preferences.

**Strengths:**

1. **Theoretical grounding**: The analysis in Section 3 provides a clean and insightful characterization of why high-reward region fidelity matters—this is a valuable contribution to the reward modeling literature.
2. **Practical relevance**: The rubric-based approach offers a data-efficient, interpretable alternative to Bradley-Terry models, especially in domains where human annotation is scarce (e.g., healthcare).
3. **Empirical rigor**: The ablation studies (Table 1, Figure 4) systematically validate the two proposed principles and show consistent gains across metrics.

**Weaknesses:**

1. The entire pipeline—initial rubric generation, top-2 selection, response scoring, and final evaluation—relies on GPT-4.1 as both proposer and judge. If GPT-4.1 systematically favors certain response styles or reasoning patterns, the refined rubric will encode and amplify this bias, leading to a self-consistent but potentially misleading reward signal.



2. The paper defines “1 Good Pair” as two responses from Gemini-2.5-Flash-Lite, and “1 Great Pair” as two from Gemini-2.5-Pro. However, sampling twice from the same model (even with temperature > 0) often yields highly similar outputs, especially on factual or constrained tasks. In such cases, the differences may be trivial (e.g., word order, minor elaboration), leading the proposer LLM to “chase noise” rather than extract meaningful quality distinctions.



3. When selecting top-2 from 16 diverse models (e.g., GPT-4o vs. Gemini-2.5-Pro), the responses often differ dramatically in length, tone, use of bullet points, emoji, hedging language, etc. While the rubric design guidelines discourage style-based criteria, the proposer LLM (GPT-4.1) may still interpret stylistic differences as quality signals (e.g., “more detailed = better”). The medical case study (Appendix I) is convincing, but it’s a single example, i hope the authors can provide broader quantitative evidence that style differences are not driving top-2 selections.


4. It remains unclear which models dominate the top-2 selections in the final iterative rounds. For instance, does GPT-5 or o1 consistently appear in the top pair? If one model (e.g., GPT-4.1 itself) dominates, the rubric may effectively become a proxy for that model’s behavior. Could the authors provide statistics on the distribution of models represented in the final top-2 pairs across all evaluated prompts?

**Questions:**

see weakness

---

> ### Author Response · Authors · 2025-11-23
> **Response to Reviewer Zt7D (1/2)**
>
> ## Bias propagation
>
> > **W1**: The entire pipeline—initial rubric generation, top-2 selection, response scoring, and final evaluation—relies on GPT-4.1 as both proposer and judge. If GPT-4.1 systematically favors certain response styles or reasoning patterns, the refined rubric will encode and amplify this bias, leading to a self-consistent but potentially misleading reward signal.
>
> **Our response**:
>
> Thank you for bringing up this important question about our decision to use GPT-4.1 both as the proposer/labeler and as the judge. We view this configuration as appropriate for the goals of our study for the following reasons:
> 1. **The current setup avoids introducing confounding features and is standard [1].** Our paper focuses on studying how off-policy responses can be used during rubric construction to mitigate reward over-optimization. To study this in a controlled manner, we follow the established setup used in prior work such as [1], treating GPT-4.1 as a stable “ground-truth’’ preference source. Using the same model for rubric proposals and preference judgments creates a clean environment where we can analyze how different training strategies affect the resulting proxy reward model. This avoids potential confounding effects by applying different models as the proposer and the judge. In practice, GPT-4.1 simply plays the role of a stand-in for whatever preference source a downstream practitioner would care about, often human annotators. To improve clarity, we have expanded our discussion of this choice in Section 4.2 in the revised manuscript. With this context, we believe our win-rate results reliably illustrate the key claim: incorporating stronger and more diverse off-policy samples makes rubric-based reward models more robust to overoptimization.
> 2. **Real improvements show in objective evaluation metrics.** We also emphasize that our conclusions do not rely solely on model-based judgments. For example, we evaluate on HealthBench [2], which is completely independent of our choice of judge: both the test prompts and the evaluation criterion are constructed independently from the training setting. The gains seen there reflect genuine improvements in model capability rather than artifacts of our evaluation design. In response to the reviewer’s concern, we have also added a new experiment in the finance domain using PRBench [3], another objective benchmark; details and full results are provided in Section 4. In particular, we observe consistent results for the finance domain, as shown in fig 7 (https://osf.io/7gt28/files/rzmjx?view_only=0e2cc586ceab4196bfcc5c4e0ac028a2). These additional evaluations offer further confirmation of the robustness and real-world utility of our method.
> 3. **Rubrics capture meaningful signals.** We have expanded our analysis of both the final rubrics and the intermediate refinements to more directly demonstrate their validity. Appendix I contains concrete examples of rubrics and refinement steps (as in the previous version), and in this revision we additionally provide high-level category summaries for each in Appendices J and K (https://osf.io/7gt28/files/kaqxn?view_only=0e2cc586ceab4196bfcc5c4e0ac028a2 and https://osf.io/7gt28/files/5zsdg?view_only=0e2cc586ceab4196bfcc5c4e0ac028a2). We provide these results so that readers can directly audit them if desired. We believe these results provide strong evidence that the constructed rubrics are meaningfully differentiating top responses rather than merely embedding noise and styles.
>
> We agree that biases may arise when the preference source and the rubric proposer differ. For instance, if humans provide preferences but LLMs help construct rubrics. We see this as an important and promising direction for future work, including techniques for automatic detecting and mitigating spurious features in rubrics. One advantage of rubric-based reward modeling is that rubrics are expressed in natural language, making them interpretable, auditable, and amenable to post-hoc edits or automated correction methods. In contrast, Bradley-Terry reward models can easily encode latent biases such as length preference [4], while offering limited transparency, as it only exposes a scalar score without interpretable intermediate structure.
>
> [1] Scaling Laws for Reward Model Overoptimization.
>
> [2] HealthBench: Evaluating Large Language Models Towards Improved Human Health
>
> [3] PRBench: Large-Scale Expert Rubrics for Evaluating High-Stakes Professional Reasoning
>
> ​​[4] RewardBench: Evaluating Reward Models for Language Modeling

---

> ### Author Response · Authors · 2025-11-23
> **Response to Reviewer Zt7D (2/2)**
>
> ## Response diversity vs. superficial style differences
>
> > **W2**: The paper defines “1 Good Pair” as two responses from Gemini-2.5-Flash-Lite, and “1 Great Pair” as two from Gemini-2.5-Pro. However, sampling twice from the same model (even with temperature > 0) often yields highly similar outputs, especially on factual or constrained tasks. In such cases, the differences may be trivial (e.g., word order, minor elaboration), leading the proposer LLM to “chase noise” rather than extract meaningful quality distinctions.
> >
> > **W3**: When selecting top-2 from 16 diverse models (e.g., GPT-4o vs. Gemini-2.5-Pro), the responses often differ dramatically in length, tone, use of bullet points, emoji, hedging language, etc. While the rubric design guidelines discourage style-based criteria, the proposer LLM (GPT-4.1) may still interpret stylistic differences as quality signals (e.g., “more detailed = better”). The medical case study (Appendix I) is convincing, but it’s a single example, i hope the authors can provide broader quantitative evidence that style differences are not driving top-2 selections.
>
> **Our response:**
>
> We agree with the limitation highlighted in Weakness 2 regarding the risk of relying on responses from a single model. This is precisely why we incorporate responses from multiple models to increase diversity (this is the 2nd rubric-construction principle). We indeed observe clear benefits from using “great and diverse’’ responses during rubric construction.
>
> For possible style pitfall by involving responses from multiple models, we argue this is not a real problem, supported by the following two points:
>
> 1. **Improvements on objective metrics.** To further demonstrate the effectiveness of our method across domains, we’ve added additional experiments on the finance domain. Details of the experiments and final results are displayed in Section 4. In particular, as shown in Fig7 (https://osf.io/7gt28/files/rzmjx?view_only=0e2cc586ceab4196bfcc5c4e0ac028a2), we get consistent results with previous tasks. . Both health and finance tasks adopt objective metrics, reinforcing that the improvements are genuine rather than stylistic.
>
> 2. **Rubrics are meaningful.** We appreciate the reviewer’s positive assessment of the medical case study. To provide more systematic evidence, we’ve included a clustering analysis of generated rubrics and show their categories in Appendix J (https://osf.io/7gt28/files/kaqxn?view_only=0e2cc586ceab4196bfcc5c4e0ac028a2). We also chase the refinement types to inspect the rubric construction process. Results are presented in Table 3 (this is also in the original version). These analyses indicate that refinements actually help to encode important criteria rather than superficial stylistic preferences. We welcome the reviewer to audit full results.
>
> ## Robustness of the “pair” construction strategy
>
> > **W4**: It remains unclear which models dominate the top-2 selections in the final iterative rounds. For instance, does GPT-5 or o1 consistently appear in the top pair? If one model (e.g., GPT-4.1 itself) dominates, the rubric may effectively become a proxy for that model’s behavior. Could the authors provide statistics on the distribution of models represented in the final top-2 pairs across all evaluated prompts?
>
> **Our response:**
>
> Thanks for raising this concern. We’ve summarized the model proportions in the final top-2 pairs and present the statistics in Appendix L. As shown in Figure 6 (https://osf.io/7gt28/files/6vqam?view_only=0e2cc586ceab4196bfcc5c4e0ac028a2), no single model dominates and there is a diverse representation of SOTA models in the final top-2 pairs.

---

### Official Review · Reviewer_W7oY · 2025-11-07

**Soundness:** 4
**Presentation:** 3
**Contribution:** 3
**Rating:** 8
**Confidence:** 2

**Summary:**

This paper addresses the critical issue of reward over-optimization in LLM post-training by proposing a rubric-based reward modeling approach. The authors first provide a theoretical analysis demonstrating that the accuracy of the reward model in the high-reward region is the key determinant of alignment performance. To tackle the sample inefficiency of obtaining such examples, they introduce an iterative workflow that refines scoring rubrics using off-policy responses from strong models. Experiments show that their method can mitigate reward over-optimization and improve model performance across diverse domains.

**Strengths:**

1. The paper provides a rigorous theoretical analysis that clearly links reward misspecification in the high-reward tail to the phenomenon of reward over-optimization.
2. The authors proposed a novel and practical iterative rubric refinement strategy, which effectively leverages off-policy data to create more robust reward signals.
3. The experiments are solid. They have conducted experiments across diverse tasks and demonstrated the effectiveness of the proposed approach. The results show clear improvements in win-rate.

**Weaknesses:**

1. The use of a simple weighted average for aggregating rubric scores may limit the expressiveness of the reward function, although the paper acknowledges that more sophisticated aggregation schemes are an area for future work.

**Questions:**

1. Have you evaluated the generalization of your per-prompt rubrics to unseen prompts?

---

> ### Author Response · Authors · 2025-11-23
> **Response to Reviewer W7oY**
>
> > **Q1**: Have you evaluated the generalization of your per-prompt rubrics to unseen prompts?
>
> Thanks for your positive feedback and encouraging words! For your question, these rubrics consistently encode shared underlying principles of good responses, such as correctness, safety, logical consistency, etc. RL training allows the model to internalize these generalizable principles rather than memorize prompt-specific rubrics. For more details, refer to Appendix J in the revised version, where we present types of rubrics with examples (https://osf.io/7gt28/files/kaqxn?view_only=0e2cc586ceab4196bfcc5c4e0ac028a2). Thanks for your insight!
>
>
> > **W1**: The use of a simple weighted average for aggregating rubric scores may limit the expressiveness of the reward function, although the paper acknowledges that more sophisticated aggregation schemes are an area for future work.
>
> Thanks for raising this point. We agree the current weighted average is simple and far from optimal. But we use this simple setup to study rubric-construction in isolation. For future directions, we can study rubric aggregation in depth. For example, violating some rubrics means critical issues and should be penalized very harshly even if all other rubrics are satisfied; in such cases, a tree-structured rubric aggregation may be more appropriate than a simple weighted average. To study this problem systematically, we can learn an aggregation function with simple parametric models, with the objective to aligning with pairwise preferences (i.e. aggregated rubric score should predict pairwise preferences).

---

### Official Review · Reviewer_aFkb · 2025-11-08

**Soundness:** 2
**Presentation:** 3
**Contribution:** 2
**Rating:** 4
**Confidence:** 4

**Summary:**

This paper investigates the problem of reward over optimization in large language model post training and identifies that the critical factor is not general reward model accuracy but its inability to reliably distinguish excellent responses from merely good ones in the high reward tail. To address this, the authors propose a rubric based reward framework that replaces traditional Bradley Terry models with structured, interpretable rubrics consisting of weighted binary criteria evaluated by another LLM. They introduce an iterative "Refinement through Differentiation" (RTD) method that progressively refines rubrics by comparing high quality and diverse off policy responses generated by stronger models. Experiments across general and healthcare domains demonstrate that focusing on these high reward regions significantly improves alignment performance boosting win rates, enhancing reward accuracy in the top quality range, and delaying reward over optimization thus validating rubric based reward modeling as a robust and explainable approach to mitigating misspecification in LLM post training.

**Strengths:**

- Clearly identifies that the core of reward over optimization lies in the ranking accuracy at the high reward tail  rather than overall error, and provides formal theoretical support.
- Proposes a complete pipeline combining rubric-based reward modeling, high-quality off-policy samples, and RTD iterative refinement, operationalizing “fine-grained differentiation in the high core region” into an automated and executable method.
- From theoretical proof of the high tail’s importance, to using stronger and more diverse responses for rubric optimization, to empirical results showing that improved high tail accuracy leads to more stable RL performance with less over-optimization.

**Weaknesses:**

- The generation and evaluation of rubrics heavily depend on stronger closed models such as GPT-4.1, essentially a “strong teacher supervising a weaker student.” It lacks systematic validation from human experts or multi-source reviews, which risks the propagation of biases from the teacher model, especially in sensitive domains such as healthcare.
- Heavy reliance on GPT-4.1 as the sole judge and labeler introduces an additional fragility: if GPT-4.1 systematically prefers the wrong answers that are the better response in certain patterns, these errors are directly baked into both the rubrics and the evaluation pipeline, and the method offers no robust mechanism to detect or correct such failure modes.
- The RTD + multi-round refinement + large-scale off-policy sampling pipeline incurs substantial computational and implementation cost; moreover, the trained model may learn to mimic rubric patterns or templates instead of genuinely improving reasoning ability. The paper lacks a systematic analysis of this potential “rubric-induced reward hacking.”
- Experiments are confined to a single model family (Qwen3-8B), a single judger GPT-4.1, and a few datasets, without comprehensive cross-scale, cross-lingual, or cross-domain validation. Furthermore, the paper does not include reasoning-intensive tasks (e.g., multi-step mathematical reasoning, code generation, or tool use), leaving it unclear whether high tail rubrics are equally effective at improving reasoning performance and mitigating over-optimization in such settings, making the conclusions more of a “strong case study” than a fully general solution.

**Questions:**

1.  Can the rubric-based rewards that have been constructed be effective in different fields (such as general, medical, and reasoning tasks)?
2. If the generation and verification of rubrics rely on closed-source models such as GPT-4.1, then how can we detect and correct any systematic errors in the evaluation model?
3. For Figure 3, the green line 4 Great Pairs appears to have a better overall trend compared to the orange line 4 Great & Diverse. It is not that 4 Great & Diverse is superior to 4 Great Pairs. Please explain.
4. Why only use one model (GPT-4.1) as the proposer instead of using different models to provide more comprehensive evidence?

---

> ### Author Response · Authors · 2025-11-23
> **Response to Reviewer aFkb (1/n)**
>
> ## Same LLM as proposer and judge
>
> >  **W1**: The generation and evaluation of rubrics heavily depend on stronger closed models such as GPT-4.1, essentially a “strong teacher supervising a weaker student.” It lacks systematic validation from human experts or multi-source reviews, which risks the propagation of biases from the teacher model, especially in sensitive domains such as healthcare.
> >
> > **W2**: Heavy reliance on GPT-4.1 as the sole judge and labeler introduces an additional fragility: if GPT-4.1 systematically prefers the wrong answers that are the better response in certain patterns, these errors are directly baked into both the rubrics and the evaluation pipeline, and the method offers no robust mechanism to detect or correct such failure modes.
> >
> > **Q2**: If the generation and verification of rubrics rely on closed-source models such as GPT-4.1, then how can we detect and correct any systematic errors in the evaluation model?
> >
> > **Q4**: Why only use one model (GPT-4.1) as the proposer instead of using different models to provide more comprehensive evidence?
>
> **Our response:**
>
> Thanks for raising this thoughtful concern regarding our use of GPT4.1 as both the proposer/labeler and the judge. We believe this setup is appropriate for our study since (1) it avoids introducing confounding factors that would obscure the effects we aim to measure, and (2) real improvements are observed on additional objective evaluation metrics. More specifically, we would like to highlight the following points:
> 1. **The current setup avoids introducing confounding features and is standard [1].** We would like to clarify that the primary goal of this paper is to study how to leverage off-policy responses in the rubric construction process to prevent reward over-optimization. To isolate this question, we treat GPT-4.1 as a gold-standard preference source we aim to align with (hence its role as the judge). We intentionally use the same model as the rubric proposer to simulate a setting in which a fixed, ground-truth reward model (GPT-4.1) provides the labels used to “train’’ a proxy (rubric-based) reward model (this setup is also used in the seminal [1] in studying reward overoptimization for Bradley-Terry based proxy reward models). This controlled design avoids introducing confounding factors arising from discrepancies between the proposer and the judge, enabling a cleaner assessment of how different rubric construction strategies influence overoptimization behavior. In practice, GPT-4.1 simply stands in for any target preference source, most notably human annotators. To better clarify and avoid potential misunderstandings, we’ve added more discussions of this design in Section 4.2 (highlighted in blue). Therefore, we believe the win-rate results are sufficient to support the main claims of the paper: leveraging stronger and more diverse off-policy responses improves the robustness of constructed rubrics against reward overoptimization.
> 2. **Real improvements show in objective evaluation metrics. To showcase the practical improvement,** we also perform objective evaluations when there exists a relevant objective benchmark. We already reported scores in Healthbench [2] (an external, objective metric that is independent of our choice of judge.)  The gains we observe on this benchmark therefore reflect genuine capability improvements rather than artifacts of our evaluation setup. To further support this point, we have added an additional experiment in the finance domain, evaluated with PRBench [3] (with details and results provided in Section 4). As shown in https://osf.io/7gt28/files/rzmjx?view_only=0e2cc586ceab4196bfcc5c4e0ac028a2, we observe consistent results across tasks and eval metrics .The metrics reported there are likewise objective. We hope these additional results offer more comprehensive evidence for the effectiveness and robustness of our method.
> 3. **Constructed rubrics chase real signals.** We also performed systematic analysis of synthetic rubrics with clustering. In the previous version, we already included clusters of rubric refinements (Table 3) with a case study (Appendix I). Additionally, we’ve included clusters of rubrics (https://osf.io/7gt28/files/kaqxn?view_only=0e2cc586ceab4196bfcc5c4e0ac028a2) in Appendix J with examples, and we also include all cluster examples in the supplement if you want to audit. See details in the general comments. These are strong evidence that the constructed rubrics are meaningfully differentiating top responses rather than merely embedding noise and styles.

---

> > ### Author Response · Authors · 2025-11-23
> > **Response to Reviewer aFkb (2/n)**
> >
> > We acknowledge this bias issue can happen if there’s a mismatch between preference source and rubric proposer. For example, when human preferences are the target but LLMs assist in rubric creation. We view this as an important and promising direction for future research, i.e., development of automated methods to identify and mitigate spurious features in rubrics. We actually think rubrics are good options in the sense of post-hoc audits and edits. **One key advantage of the rubric-based reward modeling is that rubrics in natural language are interpretable and editable. This transparency makes it significantly easier to debug and could support future automatic methods for detecting and correcting errors in the rubrics.** In contrast, Bradley-Terry reward modeling is more susceptible to encoding latent biases such as length and style bias [4] and offer limited avenues for inspection and adjustment, as it only exposes a scalar reward score without interpretable intermediate structure.
> >
> > [1] Scaling Laws for Reward Model Overoptimization.
> >
> > [2] HealthBench: Evaluating Large Language Models Towards Improved Human Health
> >
> > [3] PRBench: Large-Scale Expert Rubrics for Evaluating High-Stakes Professional Reasoning
> >
> > [4] RewardBench: Evaluating Reward Models for Language Modeling

---

> ### Author Response · Authors · 2025-11-23
> **Response to Reviewer aFkb (3/n)**
>
> ## The Scope of our paper
> > **W3**: The RTD + multi-round refinement + large-scale off-policy sampling pipeline incurs substantial computational and implementation cost; moreover, the trained model may learn to mimic rubric patterns or templates instead of genuinely improving reasoning ability. The paper lacks a systematic analysis of this potential “rubric-induced reward hacking.”
> >
> > **W4**: Experiments are confined to a single model family (Qwen3-8B), a single judger GPT-4.1, and a few datasets, without comprehensive cross-scale, cross-lingual, or cross-domain validation. Furthermore, the paper does not include reasoning-intensive tasks (e.g., multi-step mathematical reasoning, code generation, or tool use), leaving it unclear whether high tail rubrics are equally effective at improving reasoning performance and mitigating over-optimization in such settings, making the conclusions more of a “strong case study” than a fully general solution.
> >
> > **Q1**: Can the rubric-based rewards that have been constructed be effective in different fields (such as general, medical, and reasoning tasks)?
>
> **Our response:**
>
> We’d like to clarify that our paper ​​focuses on open-ended tasks that are used to be supervised by BT/generative reward models: there is no standard answer for each query. Our work does not aim to replace RLVR for reasoning tasks, and we do not position our method as a one-size-fits-all solution to post-training.
>
> A fair comparison would be between the proposed method and preference-based (BT) models or generative reward models. In this context, our method exhibits clear advantages regarding both computational cost and reward hacking.
> - For **computational cost**, our rubric construction needs only one-time cost. During RL training, since verification is not a hard task, we can apply cheaper verifiers (we use GPT-4.1-mini). On the contrary, preference-based models require iterative preference data collection+language model training, and generative reward models rely on frequent calls to strong LLM judges during training. Both alternatives are much more costly.
> - For **reward hacking**, the issue that language models learn noise/bias induced by the reward model is common for both preference-based (BT) models and generative reward models. In contrast, the rubric-based approach offers transparent, editable natural-language criteria, making it easier to audit and refine.
>
> To further illustrate the effectiveness of our method across different fields, we’ve added additional experiments on the finance field. Details and results are mentioned in Section 4 and in https://osf.io/7gt28/files/rzmjx?view_only=0e2cc586ceab4196bfcc5c4e0ac028a2, we get consistent results in different tasks with the objective metrics. We hope this can address the reviewer’s concern about cross-domain validation.
>
> Since rubrics can provide denser and more fine-grained signals, we think it is an interesting future direction to apply rubric-based supervision to reasoning tasks. While this is not straightforward, rubrics have the potential to impose more structured and nuanced supervision on reasoning traces, which may in turn lead to stronger reasoning abilities in language models. This possibility is recently supported by the Dr Tulu model by AI2: https://allenai.org/blog/dr-tulu.
>
> ## Experimental results
> > **Q3**: For Figure 4, the green line 4 Great Pairs appears to have a better overall trend compared to the orange line 4 Great & Diverse. It is not that 4 Great & Diverse is superior to 4 Great Pairs. Please explain.
>
> **Our response**:
>
> Our second rubric construction is “Effective rubric construction requires distinguishing among diverse off-policy responses.” We can elicit diverse rubric elements by increasing both the number of responses, and using different strong LLMs for generating them. Also, in https://osf.io/7gt28/files/rzmjx?view_only=0e2cc586ceab4196bfcc5c4e0ac028a2, we see that using more responses improves the performance more in objective benchmark results.

---

### Author Response · Authors · 2025-11-23
**General Response (1/2)**

We sincerely thank all reviewers for their thoughtful and constructive feedback. We are encouraged that all reviewers agree that our paper is theoretically grounded, practically impactful, and empirically rigorous. In particular, reviewers recognized:
1. The paper provides **rigorous theory** linking high-reward tail misspecification to reward over-optimization. (aFkb, W7oY, Zt7D).
2. All reviewers mentioned that the paper proposes a **novel rubric-generation method** that leverages off-policy responses to mitigate reward over-optimization.
3. Reviewers (W7oY, Zt7D, ULR5) highlighted that the paper includes **solid and comprehensive experiments**: the empirical studies span multiple domains, pairing conditions, and refinement types, and the ablations clearly validate the contribution of each component in the proposed approach.

The major concern raised by several reviewers is that the proposed rubric-construction process may chase noise instead of meaningful signals, leading the final rubrics to encode superficial stylistic criteria instead of fundamental rules. This concern stems from two sources: (1) using GPT-4.1 as both the proposer and the judge, and (2) questions about response diversity and the actual refinements. To address these points, we’ve conducted additional experiments and revised the paper to improve the clarity and completeness of our manuscript. New figures and tables are added to the latest version and also in our responses for reviewers’ convenience. We summarize our justification and revision as follows:
1. **Using the same model as proposer and judge follows standard practice [1] and avoids introducing confounding factors.** The primary goal of this paper is to study how to leverage off-policy responses in the rubric construction to mitigate reward over-optimization. To isolate this question, we treat GPT-4.1 as a gold-standard preference source we aim to align with (hence its role as the judge). We intentionally use the same model as the rubric proposer to simulate a setting where a fixed, ground-truth reward model (GPT-4.1) provides the labels used to “train’’ a proxy (rubric-based) reward model. This setup is standard and also adopted in [1] in studying reward overoptimization for Bradley-Terry based proxy reward models. This controlled design avoids confounding factors from proposer-judge mismatches, allowing a cleaner assessment of how different rubric construction strategies influence over-optimization. In practice, GPT-4.1 simply stand-in for any target preference source, most notably human annotators.
- Revision: We’ve added more discussions of this design in Section 4.2. Specifically, we mention:
> To isolate the effect of different rubric refinement strategies on reward over-optimization, we adopt the synthetic oracle setting proposed by the seminal work Gao et al. (2023). In this framework, a strong LLM acts as a proxy for the “gold-standard” preference source. This oracle model serves a dual purpose: it generates the rubrics data used for reward modeling and serves as the final judge for performance evaluation. By unifying the annotation and evaluation sources, this design eliminates confounding factors arising from preference mismatch between the training data and the judge. This allows us to clearly assess how effectively different rubric refinement strategies mitigate reward over-optimization.

We thank the reviewer for raising this concern and we hope the revision can better clarify the goal of our paper and avoid potential misunderstandings.

2. **Real improvements show in objective evaluation metrics.** We also emphasize that our conclusions do not rely solely on model-based judgments. We perform evaluations using objective benchmarks where applicable. For example, we evaluate on HealthBench [2], which is completely independent of our choice of judge. The gains seen there reflect genuine improvements in model capability rather than artifacts of our evaluation design. In response to the reviewer’s concern, we’ve added additional experiments on another task with objective evaluation metrics. We hope these additional evaluations further validate robustness and real-world utility of our method.
- Revision: We have added a new task in the finance domain using PRBench [3], another objective benchmark. Details of the experiments and full results have been added in Section 4 in the revised manuscript. See https://osf.io/7gt28/files/rzmjx?view_only=0e2cc586ceab4196bfcc5c4e0ac028a2. By characterizing the training dynamics with both oracle preference and objective metrics, we clearly show the benefits of leveraging great and diverse responses in rubric constructions.

---

> ### Author Response · Authors · 2025-11-23
> **General Response (2/2)**
>
> 3. **Rubrics capture meaningful signals instead of superficial stylistic preference.** In the previous version, we provided concrete examples of rubrics and refinements (see Table 3 and Appendix I). In response to reviewers’ common concerns, we have expanded our analysis of the final rubrics and refinements to more directly demonstrate their validity. We believe these results provide strong evidence that the constructed rubrics are meaningfully differentiating top responses rather than merely capturing noise or stylistic variation. We welcome reviewers to audit these results in the revised manuscript.
> - Revision: We’ve conducted principled clustering analyses of both the final rubrics and the refinements during rubric constructions. High-level category summaries are presented in https://osf.io/7gt28/files/kaqxn?view_only=0e2cc586ceab4196bfcc5c4e0ac028a2 and https://osf.io/7gt28/files/5zsdg?view_only=0e2cc586ceab4196bfcc5c4e0ac028a2 (Appendices J and K).
> In addition, to further rule out the possibility that the rubrics reflect the preferences of only a subset of SOTA models, as suggested by the reviewer ULR5, we’ve summarized the model proportion in the last-round top-2 pairs. As shown in Figure 6 (https://osf.io/7gt28/files/6vqam?view_only=0e2cc586ceab4196bfcc5c4e0ac028a2) in Appendix L, no single model dominates and there is a diverse representation of SOTA models in the final top-2 pairs.
>
> All revisions are highlighted in blue in the latest manuscript.
>
> [1] Scaling Laws for Reward Model Overoptimization.
>
> [2] HealthBench: Evaluating Large Language Models Towards Improved Human Health
>
> [3] PRBench: Large-Scale Expert Rubrics for Evaluating High-Stakes Professional Reasoning

---

### Meta-Review · Area_Chair_MHCt · 2026-01-04

**Summary:**

Although the review scores are borderline, the rebuttal addressed the main concerns (proposer/judge confounds, diversity/style pitfalls) to some extent, and there is value in accepting this paper.

**Reviewer Concerns:**

The authors addressed the main reviewer concerns (proposer/judge confounds, diversity/style pitfalls) while further strengthening claims with gains on objective benchmarks with HealthBench and PRBench-Finance. The rebuttal did not fully address the concerns (as mentioned by Reviewer ULR5). The authors can further add experiments to demonstrate the method generalizes to other judges or human preferences.

**Reviewer Scores:**

Scores are 4846, while the reviewer with score 8 has a lower confidence. Thus this is a borderline paper. Reviewer ULR5 (with score 6) mentioned the rebuttal did not address their concerns and the scores may not have changed even with a full discussion period.

---

### Decision · Program_Chairs · 2026-01-26

Accept (Poster)